# The carboxyl-terminal sequence of bim enables bax activation and killing of unprimed cells

Xiaoke Chi[1,2†], Dang Nguyen[2,3†], James M Pemberton[2,3†], Elizabeth J Osterlund[2,4], Qian Liu[2], Hetal Brahmbhatt[1,2,5], Zhi Zhang[6,7], Jialing Lin[6,7], Brian Leber[5], David W Andrews[2,3,4*]

[1]Department of Chemistry and Chemical Biology, McMaster University, Hamilton, Canada; [2]Biological Sciences, Sunnybrook Research Institute, Toronto, Canada; [3]Department of Medical Biophysics, University of Toronto, Ontario, Canada; [4]Department of Biochemistry, University of Toronto, Toronto, Canada; [5]Department of Medicine, McMaster University, Hamilton, Canada; [6]Department of Biochemistry, University of Oklahoma Health Sciences Center, Oklahoma City, United States; [7]Molecular Biology and Stephenson Cancer Center, University of Oklahoma Health Sciences Center, Oklahoma City, United States

**Abstract** The Bcl-2 family BH3 protein Bim promotes apoptosis at mitochondria by activating the pore-forming proteins Bax and Bak and by inhibiting the anti-apoptotic proteins Bcl-XL, Bcl-2 and Mcl-1. Bim binds to these proteins via its BH3 domain and to the mitochondrial membrane by a carboxyl-terminal sequence (CTS). In cells killed by Bim, the expression of a Bim mutant in which the CTS was deleted (BimL-dCTS) triggered apoptosis that correlated with inhibition of anti-apoptotic proteins being sufficient to permeabilize mitochondria isolated from the same cells. Detailed analysis of the molecular mechanism demonstrated that BimL-dCTS inhibited Bcl-XL but did not activate Bax. Examination of additional point mutants unexpectedly revealed that the CTS of Bim directly interacts with Bax, is required for physiological concentrations of Bim to activate Bax and that different residues in the CTS enable Bax activation and binding to membranes.

**\*For correspondence:**
david.andrews@sunnybrook.ca

[†]These authors contributed equally to this work

**Competing interests:** The authors declare that no competing interests exist.

## Introduction

Apoptosis is a highly conserved form of programmed cell death that can be triggered by extrinsic or intrinsic signals. It plays a fundamental role in maintaining homeostasis by eliminating old, excessive or dysfunctional cells in multicellular organisms (*Kerr et al., 1972*). Defective regulation of apoptosis has been found in many diseases (*Favaloro et al., 2012*) and is considered one of the hallmarks of cancer (*Hanahan and Weinberg, 2011*).

Bcl-2 family proteins play a decisive role in apoptosis initiated by intrinsic signaling by regulating the integrity of the mitochondrial outer membrane (MOM). Commitment to apoptosis is generally regarded as due to MOM permeabilization (MOMP) releasing cytochrome c and pro-apoptotic factors from the intermembrane space into the cytoplasm. These factors activate the executioner caspases that mediate cell death (*Chipuk et al., 2006*). Direct interactions between Bcl-2 family proteins govern both initiation and inhibition of MOMP (*Kale and Osterlund, 2017*). The Bcl-2 family of proteins that regulate apoptosis includes the anti-apoptotic proteins Bcl-XL, Bcl-2 and Mcl-1 that inhibit the process and share four Bcl-2 homology domains. These homology domains, referred to as BH domains, are also shared by the pro-apoptotic proteins Bax and Bak that permeabilize the MOM directly. Both pro- and anti-apoptotic multi-domain Bcl-2 family proteins are regulated by

direct binding interactions with a group of proteins including Bim, Bid, Puma, Hrk, Bad and Noxa that contain a single region of homology, the Bcl-2 homology domain number 3, and are therefore referred to collectively as BH3-proteins. These proteins promote apoptosis by releasing sequestered activated Bax, Bak and BH3-proteins that activate Bax and Bak from one or more of the anti-apoptotic proteins. The subset of BH3 proteins that bind to and activate Bax or Bak include Bid, Bim and Puma (*Chi et al., 2014*). Thus far, the biochemical basis for the differences between BH3-proteins that inhibit anti-apoptotic proteins and those that activate Bax and Bak has been attributed entirely to differences in affinities of the BH3-domain for the BH3-peptide binding sites on multi-domain pro- and anti-apoptotic proteins. However, static affinities and variations in expression levels permit only coarse regulation of cell death. Changes in the equilibrium binding of Bcl-2 family proteins on the MOM enable finer control. For example, at physiologic concentrations, the BH3-protein Bid only activates Bax after Bid has bound to a membrane and undergone a specific conformational change (*Lovell et al., 2008*; *Shamas-Din et al., 2013a*). Binding to membranes also enables interaction of Bid with MTCH2 on the MOM to greatly accelerate the Bid conformational change that results in Bax activation (*Shamas-Din et al., 2013a*). However, it remains unclear whether membrane interactions by other BH3-proteins like Bim contribute to Bax activation.

The BH3-protein Bim is an important mediator of apoptosis initiated by many intracellular stressors (*Concannon et al., 2010*; *Mahajan et al., 2014*; *Puthalakath et al., 2007*). Three major isoforms of Bim result from alternative mRNA splicing: BimEL, BimL, and BimS (*O'Connor et al., 1998*). All three isoforms include the BH3-domain required for binding other Bcl-2 family proteins, and a C-terminal sequence (CTS) that binds the protein to the MOM (*Wilfling et al., 2012*). BimEL and BimL also share a dynein light chain binding motif (LC1) that sequesters these isoforms at the cytoskeleton (*Lei and Davis, 2003*). However, recent evidence also suggests that the LC1 sequence mediates Bim oligomerization by binding to the DLC1 protein (*Singh et al., 2017*). Isoform BimS does not contain the LC1-binding motif (*Lei and Davis, 2003*), and is rarely present in healthy cells, while BimL and BimEL are present in most tissue types (*O'Reilly et al., 2000*). Bim has a particularly important function as a regulator of anti-apoptotic proteins, as it binds and thereby inhibits by mutual sequestration all known anti-apoptotic proteins (*Chen et al., 2005*; *Shamas-Din et al., 2013b*). Until recently, it was unknown why Bim binds to Bcl-XL with sufficient affinity to resist displacement by small molecule BH3-mimetics, while other BH3-proteins, such as Bad, are displaced (*Aranovich et al., 2012*). In addition to interactions via the BH3-domain, residues within the Bim CTS bind to Bcl-XL, and thereby increase the affinity of the interaction by 'double-bolt locking' providing an explanation for the observations with BH3 mimetic drugs (*Liu et al., 2019*). Here, we investigated whether the CTS of Bim also contributes to the functional and physical interactions between Bim and Bax.

We demonstrate that both primary cells and cell lines have a range of apoptotic responses to the expression of a truncated BimL protein lacking the CTS (BimL-dCTS), while expression of full-length BimL was sufficient to kill all of these cells. To determine the molecular mechanism that underlies this difference, the two pro-apoptotic functions of Bim; activation of Bax and inhibition of Bcl-XL, were quantified using purified full-length BimL and BimL mutant proteins and cell-free assays. Replacing the CTS of Bim with an alternative tail-anchor that binds the protein to mitochondrial membranes did not fully restore Bax activation function, demonstrating that sequences within the Bim CTS rather than membrane binding contribute to Bax activation. Site-directed mutagenesis of the Bim CTS also revealed residues important for binding to membranes that were not required for Bax activation (e.g. I125). Furthermore, specific residues within the CTS were identified that are required for BimL to efficiently activate Bax, but that are not required for BimL to bind to and inhibit Bcl-XL. Evidence in cell free assays demonstrated that BimL CTS residues L129 and I132 physically interact with Bax in the BH3 binding groove and are required for Bim to activate Bax. Together, our data demonstrates that the unusual sequence of the CTS of Bim separately controls both membrane binding and Bax activation.

## Results

### The CTS of Bim variably contributes to the pro-apoptotic activity of Bim in different cell lines

Removing the CTS from Bim abrogates pro-apoptotic activity in HEK293 cells (*Weber et al., 2007*). While this observation has generally been ascribed to loss of binding of Bim to MOM our observation that the CTS is also involved in binding BimEL to Bcl-XL (*Liu et al., 2019*) suggested that there may be other explanations for the loss of pro-apoptotic activity for Bim when the CTS is removed. To determine the contribution of the Bim CTS to pro-apoptotic activity, a BimL mutant was generated in which the previously characterized membrane binding domain (carboxyl-terminal residues P121- H140) were deleted (BimL-dCTS) (*Wilfling et al., 2012*; *Liu et al., 2019*). This mutant was expressed in cells and the effectiveness of induction of cell death was compared to expression of full-length BimL by confocal microscopy. To detect expression of the constructs in live cells, they included an N-terminally fused Venus fluorescent protein (indicated by a superscripted v in the name). Thus, a construct in which Venus was fused to the amino-terminus of BimL is referred to here as $^V$BimL while the mutant lacking the CTS is $^V$BimL-dCTS. As an inactive control, we used $^V$BimL-4E a mutant in which four conserved hydrophobic residues in the BH3-domain of BimL were replaced with glutamate, thereby preventing binding to all other multi-BH domain Bcl-2 family proteins (*Chen et al., 2005*; *Liu et al., 2019*).

To assay pro-apoptotic activity, the constructs were expressed in primary cells and cell lines and both expression and cell death were measured using confocal microscopy. Apoptosis was assessed by detecting externalization of phosphatidylserine by Annexin V staining in cells expressing detectable levels of $^V$BimL or the $^V$BimL mutants as measured by Venus fluorescence. As a positive control for activation of Bax $^V$tBid, the activated form of the BH3-protein Bid fused to the C-terminus of the Venus fluorescent protein was also expressed in cells. As expected, expression of $^V$BimL induced apoptosis in all cell types tested, while the negative control protein $^V$BimL-4E did not (*Figure 1A*). Similarly, $^V$tBid induced apoptosis in all the cell types except HEK293 cells. As reported previously for Bim-dCTS, the fluorescent version ($^V$BimL-dCTS) failed to induce cell death in HEK293 cells (*Weber et al., 2007*). In contrast, expression of $^V$BimL-dCTS induced apoptosis to levels similar to $^V$BimL in HCT116, BMK and MEF cells but may have reduced potency in CAMA-1 cells. Comparing the AnnexinV intensities for individual cells at a variety of equivalent expression levels of the Bim mutants across the different cell types revealed that the CTS of Bim was required for the pro-apoptotic activity of Bim in HEK293 cells (*Figure 1—figure supplement 1*).

To determine if this difference in response to $^V$BimL-dCTS expression is a function of the extent to which the apoptotic machinery is loaded in MOM, mitochondria were purified from cells resistant (HEK293) and sensitive (MEF) to $^V$BimL-dCTS expression and assayed by BH3-profiling (*Potter and Letai, 2016*). This assay measures loading of anti-apoptotic proteins with BH3-proteins or active Bax/Bak. Unlike BH3-profiling experiments conducted with BH3-peptides, in these experiments purified full-length proteins were used. Thus, purified cBid, BimL, BimL-dCTS, Bad and Noxa proteins were incubated with mitochondria from each of the cell lines and MOMP was measured by separating supernatant and pellet fractions for each reaction, and immunoblotting for cytochrome c released from the intermembrane space as previously described (*Pogmore et al., 2016*). Immunoblots were quantified and MOMP assessed as % cytochrome c released (*Figure 1B*). As expected from the data in *Figure 1A*, addition of recombinant BimL was sufficient to induce cytochrome c release from mitochondria from both HEK293 and MEF cells. However, addition of BimL-dCTS induced cytochrome c release only in the MEF mitochondria confirming that resistance to BimL-dCTS in HEK293 cells is manifest at mitochondria. Mitochondria purified from HEK293 cells were more sensitive to BimL protein than to recombinant cBid, a phenomenon that is also seen when $^V$tBid is expressed in these cells (*Figure 1A*). This result may be due to the inherent differences between Bim and Bid reported previously (*Sarosiek et al., 2013*). Nevertheless, in HEK293 cells, $^V$tBid was more active than BimL-dCTS and functionally equivalent to $^V$BimL in every other cell line tested.

One potential explanation for the difference in response to BimL-dCTS and BimL is that the mitochondria in the cell lines have different dependencies on multi-domain anti-apoptotic proteins for survival, a phenomenon known as priming. If BimL-dCTS has lost one of the functions of Bim such as activating Bax or Bak or inhibiting one of the multi-domain anti-apoptotic proteins Bcl-2, Bcl-XL and

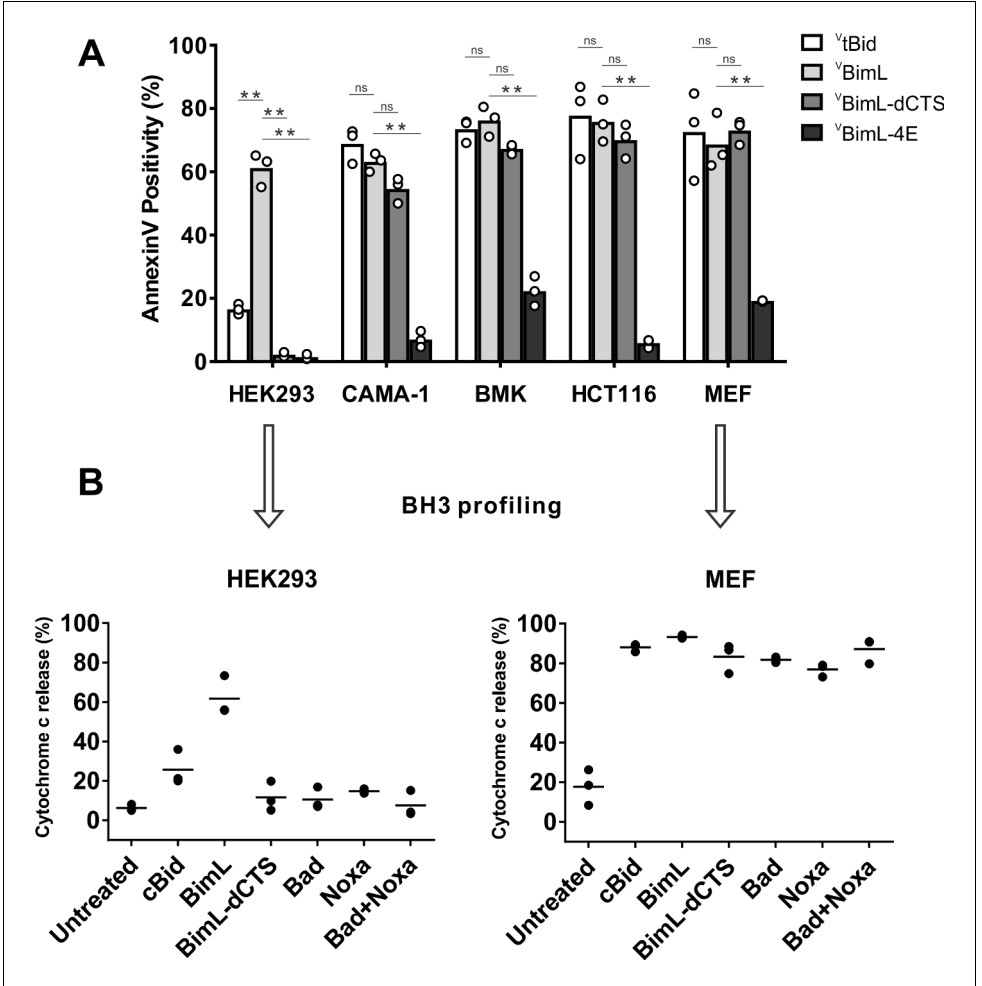

**Figure 1.** Cell lines demonstrate a range of apoptotic response to BimL-dCTS expression. (**A**) Venus, VtBid VBimL, VBimL-dCTS or VBim-4E were expressed in the indicated cell lines by transient transfection. The cells were stained with the nuclear dye Draq5 and rhodamine labeled Annexin V, and imaged by confocal microscopy to identify cells undergoing apoptosis. At least 400 cells were analyzed for each condition. The Y axis indicates the percentage of Venus-positive cells that stained positive with Annexin V. Open circles represent the average for each replicate, while the bar height, represents the average for all three replicates. The means were assessed for significant differences using a one-way ANOVA within each group followed by a Tukey's multiple comparisons test. *p-values<0.05, **p-<than 0.01, ns are non-significant p-values (>0.05). (**B**) BH3 profiling of mitochondria isolated from HEK293 and MEF cells. Mitochondria (1 mg/mL) were incubated with 500 nM of the indicated recombinant BH3-protein(s) for 1 hr at 37˚C. Cytochrome c release, indicative of MOMP, was quantified by immunoblotting. Data from three independent experiments are shown as individual points, with lines representing the average. Some dots are not visible due to overlap.

The online version of this article includes the following source data and figure supplement(s) for figure 1:

**Source data 1.** Source data indicating that BimL requires a CTS to induce MOMP in HEK293 cells.

**Source data 2.** Image data for MEF mitochondrial priming experiments.

**Source data 3.** Image data for HEK293 mitochondrial priming experiments.

**Figure supplement 1.** Correlation between expression of VBH3-proteins and apoptosis measured as Annexin V labeling.

**Figure supplement 1—source data 1.** Source data of binned Venus intensities and Annexin V positivity.

---

Mcl-1 it would be expected to have different activities on mitochondria with different priming. Therefore, to better understand why BimL-dCTS can only permeabilize MEF mitochondria and not mitochondria from HEK293 cells, we compared the sensitivity of mitochondria from the two cell types to addition of BH3-proteins Bad or Noxa that inhibits Bcl-2 and Bcl-XL or Mcl-1, respectively,

but that do not activate Bax or Bak (*Kale and Osterlund, 2017*). Incubation of full-length Bad and/or Noxa with mitochondria from HEK293 cells failed to induce cytochrome c release, while the addition of Noxa or Bad was sufficient to permeabilize MEF mitochondria (*Figure 1B*). This data suggests that HEK293 cells not depend on expression of Bcl-2, Bcl-XL or Mcl-1 sequestering active Bax, Bak or their BH3-activators while mitochondria from MEFs depend on expression of Mcl-1 and Bcl-XL to prevent apoptosis (*Lessene et al., 2013*). The results further suggest that removal of the CTS from BimL results in a mutant protein that only kills cells dependent on one or more multi-domain anti-apoptotic proteins for survival. That BimL-dCTS does not kill HEK293 cells further suggests that it does not activate sufficient Bax or Bak to overcome the unoccupied anti-apoptotic proteins in this cell line. In this way, BimL-dCTS functions as a sensitizer similar to proteins like Bad and Noxa. However, unlike other relatively specific sensitizer proteins, the BH3-region of BimL-dCTS binds to and inhibits Bcl-2, Bcl-XL and Mcl-1. Indeed, we have shown that in live cells BimEL-dCTS binds to and inhibits Bcl-2 and Bcl-XL but is more easily displaced than BimEL by small molecule BH3 mimetics (*Liu et al., 2019*).

## Full-length BimL is required to kill cultures of primary cortical neurons

Our data with cell lines and their respective purified mitochondria suggests that BimL-dCTS does not kill cells that do not depend on anti-apoptotic proteins for survival. To test this in a more biologically relevant system, we cultured primary murine cortical neurons and assayed their response to expression of the BimL mutants. To enable regulated expression in primary cortical neurons the coding regions for $^V$BimL, $^V$BimL-4E, and $^V$BimL-dCTS were cloned into a tetracycline-responsive lentiviral vector, and introduced into primary cortical neuron cultures through lentiviral infection. After culture for 16 days in vitro, BimL expression was induced in the neurons by the addition of doxycycline. Neuronal cell death was assayed using confocal microscopy after staining neurons with TMRE, a dye that only accumulates in active mitochondria. Thus, a lack of TMRE dye accumulation (TMRE negative) indicates loss of mitochondrial transmembrane potential and in response to expression of a BH3-protein is an early indication of commitment to cell death. Quantification of Venus-expressing neuronal cell bodies revealed that as expected $^V$tBid and $^V$BimL expression killed cultured primary neurons while $^V$BimL-4E did not (*Figure 2A–B*). However, the expression of $^V$BimL-dCTS was largely ineffective to induce cell death in cultured primary cortical neurons (*Figure 2B*). Our data is consistent with previous reports suggesting that primary murine cultures of hippocampal neurons become resistant to induction of apoptosis by external stimuli over time in culture. This resistance has been reported to be due to a difference in Bcl-2 family protein expression that results in decreased mitochondrial 'priming', explaining why our cultures of primary cortical neurons are resistant to $^V$BimL-dCTS (*Sarosiek et al., 2017*).

To determine if resistance to induction of cell death by BimL-dCTS is due to differential sensitivity of neuronal mitochondria to induction of MOMP by BimL and BimL-dCTS, mitochondria were isolated from embryonic day 15 (E15) mouse brains, the same age used to culture primary cortical neurons. Brain mitochondria were used instead of isolating mitochondria from neuronal cultures due to the low yield from primary cultured neurons. Untreated mitochondria from day E15 brain released only low levels of cytochrome c. As expected, addition of 0.1 nM recombinant BimL was sufficient to elicit MOMP as measured by cytochrome c release and detection in the supernatant. In contrast, 100 times more BimL-dCTS (10 nM) failed to induce MOMP (*Figure 2C*).

Taken together our data suggest that BimL-dCTS kills cells in which the mitochondria are sensitive to inhibition of anti-apoptotic proteins by sensitizers such as Bad and Noxa. Thus, BimL-dCTS did not permeabilize mitochondria extracted from HEK293 cells or E15 whole murine brains, and as a result, BimL-dCTS expression did not kill HEK293 cells or primary cultures of cortical neurons. This finding suggests that inhibition of anti-apoptotic proteins is not sufficient to kill these cells. Therefore, BimL-dCTS differs mechanistically from BimL as the latter kills both cell types resistant and sensitive to BimL-dCTS. Compared to BimL, BimL-dCTS is missing the membrane-binding domain and therefore is not expected to localize at mitochondria (*Liu et al., 2019*); however, the relationship between Bim binding to membranes and Bim-mediated Bax activation has not been extensively studied. To determine how the molecular mechanism of BimL-dCTS differs from BimL the activities of the proteins were analyzed using cell-free assays.

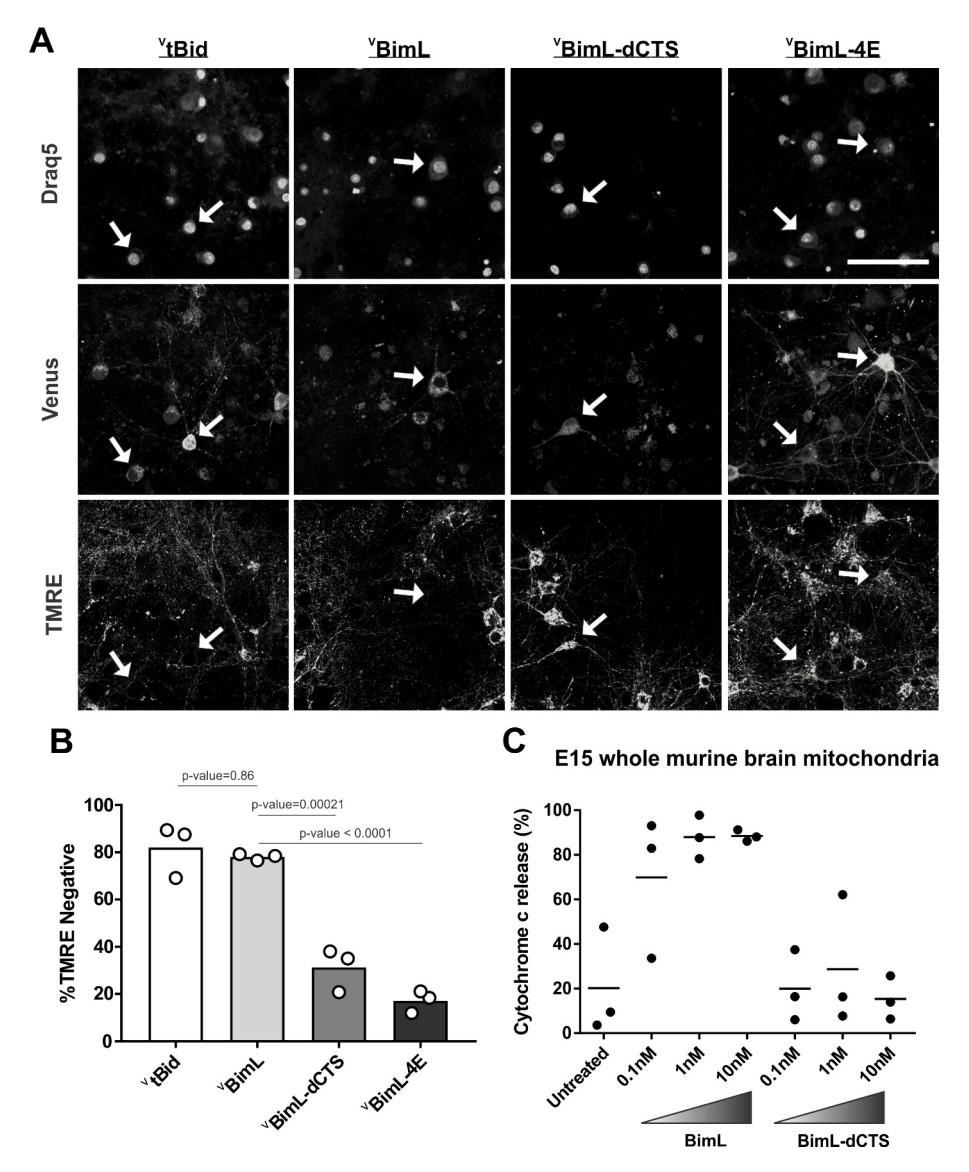

**Figure 2.** Full-length BimL is required to kill cultures of primary cortical neurons. (A) Representative images of primary cortical neurons infected with lentivirus to express ᵛtBid, ᵛBimL, ᵛBimL-dCTS or ᵛBimL-4E as indicated above. Each row is a different channel as indicated to the left, for the same field of cells. White arrows indicate illustrative neurons expressing Venus and hence fluorescent. Scale bar is 80 μm. (B) Quantified data from Venus expressing primary cortical neurons. Percentage of Venus-expressing cells that stain negative with TMRE dye (% TMRE Negative). Open circles; averages for three biological replicates each representing 90–1000 cells analyzed. The Bar height; mean. A one-way ANOVA was used followed by a Tukey's multiple comparisons test to compare the means of each group. (C) Mitochondria extracted from embryonic day 15 (E15) mouse brains (0.5 mg/mL) were incubated with the indicated BH3-only proteins. Cytochrome c release, indicative of MOMP, was quantified using immunoblotting. Each point (black circle) represents one independent replicate, with the line representing the average across all three.

The online version of this article includes the following source data for figure 2:

**Source data 1.** Source data for experiments demonstrating that ᵛBimL-dCTS does not kill mature cultures of mouse neurons.

**Source data 2.** Image data for E15 brain mitochondria priming experiments.

## The Bim CTS mediates BimL binding to both Bax and membranes

To investigate the pro-apoptotic mechanism of BimL and BimL-dCTS without interference from other cellular components, both were purified as full-length recombinant proteins and assayed using liposomes and/or isolated mitochondria. To measure direct-activation of Bax by Bim, either BimL or BimL-dCTS was incubated with recombinant Bax and liposomes encapsulating the dye and quencher pair: ANTS (8-Aminonaphthalene-1,3,6-Trisulfonic Acid, Disodium Salt) and DPX (p-Xylene-Bis-Pyridinium Bromide). In this well-established assay (*Chi et al., 2014*), increasing amounts of BimL activated Bax results in membrane permeabilization measured as an increase in fluorescence due to the release and separation of encapsulated dye and quencher (*Figure 3A*). This result is consistent with previous observations that picomolar concentrations of BimL induce Bax-mediated membrane permeabilization (*Sarosiek et al., 2013*). In contrast, three orders of magnitude higher concentrations of BimL-dCTS were required to induce Bax-mediated liposome permeabilization (*Figure 3A*), suggesting that either or both of binding to membranes and the specific CTS of Bim are required for efficient Bax activation. As expected, similar results were obtained for Bax-mediated release of mitochondrial intermembrane space proteins (*Figure 3B*). For these experiments, MOMP was measured as release of the fluorescent protein mCherry fused to the N-terminal mitochondrial import signal of SMAC (SMAC-mCherry) from the intermembrane space of mitochondria (*Shamas-Din et al., 2014*).

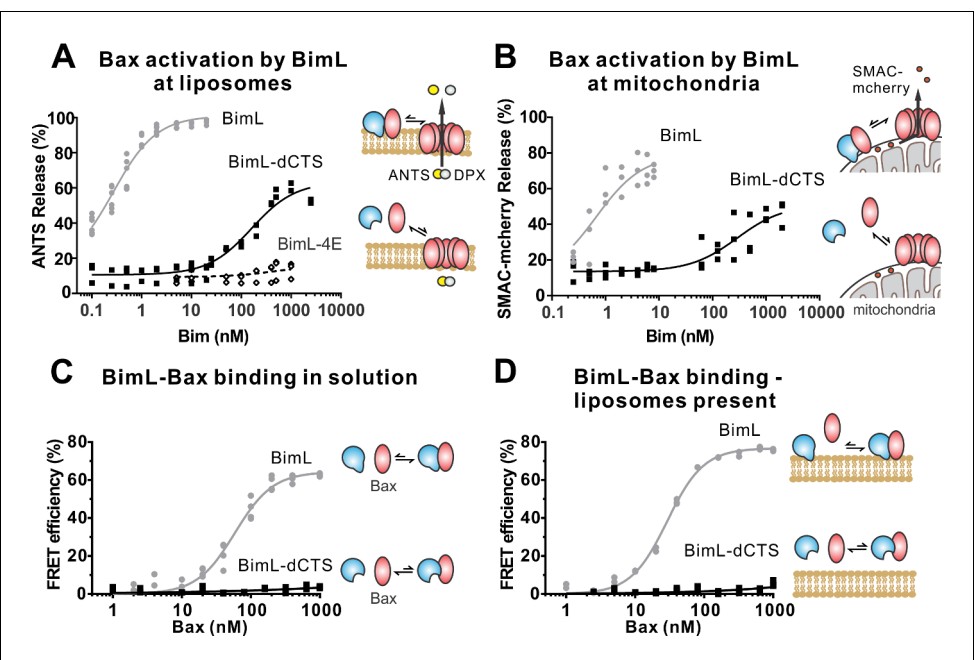

**Figure 3.** The Bim CTS is required to activate Bax to permeabilize membranes. Cartoons indicate the binding interactions being measured. Equilibria symbols indicate the predicted balance of complexes for BimL (blue), Bax (red), liposomes (tan), mitochondria (black). For each graph, data from three independent experiments are shown as individual points. Due to overlap, some points may not be visible. (A) Activation of Bax by BimL assessed by measuring permeabilization of ANTS/DPX filled liposomes (0.04 mg/mL) after incubation of Bax (100 nM) with the indicated concentrations of BimL or BimL-dCTS. Fluorescence intensity, indicative of membrane permeabilization, was measured using the Tecan infinite M1000 microplate reader and converted to percent release by comparison with detergent-mediated liposome lysis. (B)Permeabilization of the outer mitochondrial membrane by Bax (50 nM) in response to activation by the indicated amounts of Bim and BimL-dCTS was assessed by measuring SMAC-mCherry release from mitochondria. (C) Bim binding to Bax in solution measured by FRET. Alexa568-labeled BimL or BimL-dCTS (4 nM) was incubated with the indicated amounts of Alexa647-labeled Bax and FRET was measured from the decrease in Alexa568 fluorescence. (D) Bim binding to Bax measured by FRET in samples containing mitochondrial-like liposomes. FRET was measured as in (C) with 4 nM Alexa568-labeled BimL or BimL-dCTS and the indicated amounts of Alexa647-labeled Bax.

The online version of this article includes the following source data for figure 3:

**Source data 1.** Source data for experiments demonstrating that BimL-dCTS cannot bind to nor activate Bax.

Similar to the results with liposomes (*Figure 3A*), and mitochondria from cell lines (*Figures 1–2*) BimL but not BimL-dCTS triggered Bax mediated SMAC-mCherry release from mitochondria isolated from Bax - /- Bak-/-cells (*Figure 3B*). In experiments with liposomes and mitochondria, very small amounts of Bim were sufficient to trigger membrane permeabilization because once activated, Bax recruits and activates additional Bax molecules (*Tan et al., 2006*). To assess the impact of the Bim CTS on the interaction between Bim and Bax, binding was measured using Förster resonance energy transfer (FRET). For these experiments, recombinant BimL proteins were labeled with the donor fluorophore Alexa568, while Bax was labeled with the acceptor fluorophore Alexa647. Unexpectedly, and unlike the BH3-only protein tBid (*Lovell et al., 2008*), BimL bound to Bax even in the absence of membranes (*Figure 3C*), while BimL-dCTS had no relevant Bax binding in the presence or absence of mitochondrial-like liposomes (*Figure 3C–D*). Binding of Bim to Bax in solution suggests that the CTS of Bim may be directly involved in Bim-Bax heterodimerization independent of Bim binding to membranes.

To confirm in our system that the labeled BimL proteins bind to membranes via the CTS sequence, binding of Alexa568-labeled recombinant BimL and BimL-dCTS to DiD labeled liposomes was measured by FRET (*Figure 4A*). In these experiments, DiD serves as an acceptor for energy transfer from Alexa568-labeled BimL. The same approach was used to quantify BimL binding to mitochondrial outer membranes with mitochondria isolated from BAK-/-mouse liver (*Figure 4B*), which lack Bax and Bak (*Shamas-Din et al., 2013a*). In both cases, BimL spontaneously bound to

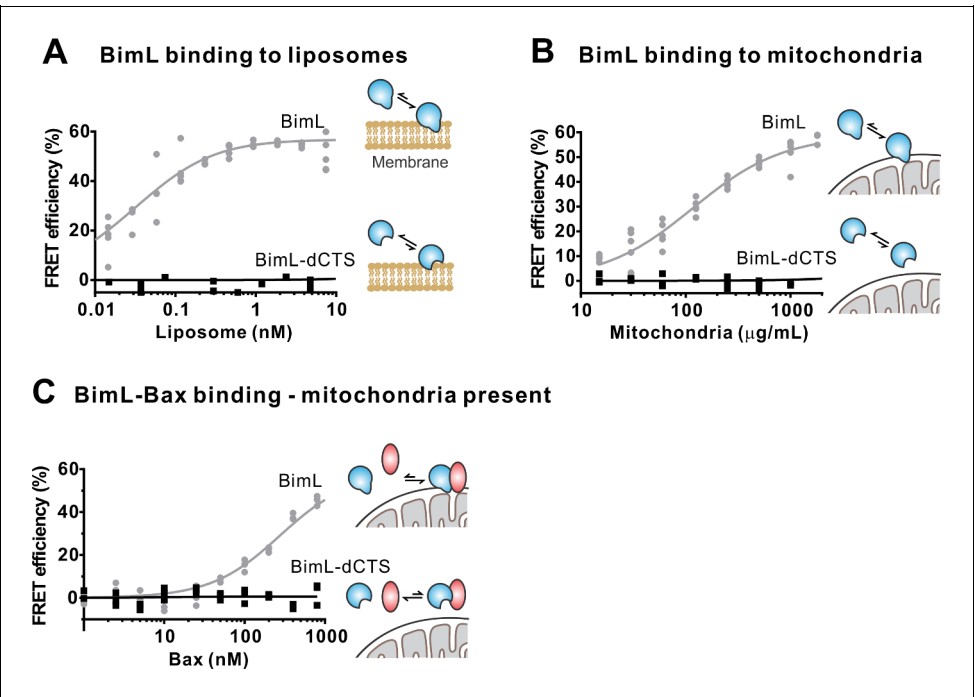

**Figure 4.** The Bim CTS is required to bind BimL to membranes in vitro. Cartoons indicate the binding interactions being measured. Equilibria symbols indicate the predicted balance of complexes for Bim (blue), Bax (red), liposomes (tan), and mitochondria (black). Unless stated otherwise, for each graph, data from three independent experiments are shown as individual points. Due to overlap, some points may not be visible. (A) The CTS of Bim is necessary for Bim to bind to liposomes. Bim binding to mitochondrial-like liposomes assessed by measuring FRET between 20 nM Alexa568-labeled BimL or BimL-dCTS and the indicated amounts of DiD-labeled liposomes. (B) The CTS of Bim is necessary for Bim to bind to mitochondria. Binding of 4 nM Alexa568-labeled BimL (n = 5) or BimL-dCTS to the indicated amounts of DiD-labeled mouse liver mitochondria was assessed by measuring FRET. (C) The CTS of Bim is necessary for Bim to bind to Bax at mitochondria. Bim binding to Bax was measured by FRET in samples containing mouse liver mitochondria, 4 nM Alexa568-labeled BimL (gray) or BimL-dCTS (black) and the indicated amounts of Alexa647-labeled Bax.

The online version of this article includes the following source data for figure 4:

**Source data 1.** Source data for experiments demonstrating the Bim CTS is required to bind BimL to membranes.

membranes with picomolar affinity, while stable binding of BimL-dCTS to liposomes and mitochondria was not-detectable (*Figure 4A–B*). Furthermore, BimL-dCTS again had no relevant binding to Bax even in the presence of purified mitochondria (*Figure 4C*).

Taken together, our data strongly suggest that the CTS of Bim is required for both BimL to bind to membranes in vitro and for binding Bax with or without membranes. Alternatively purified BimL-dCTS may be completely non-functional. To demonstrate that purified BimL-dCTS binds to and inhibits Bcl-XL as shown for ^VBimL-dCTS expressed in cells (*Figure 1*) and in *Liu et al. (2019)*, inhibition of Bcl-XL was measured using liposomes and mitochondria.

## The CTS is not required for BimL to inhibit Bcl-XL

In addition to direct Bax activation, Bim promotes apoptosis by binding to Bcl-XL and displacing either activator BH3-proteins (Mode 1) or activated Bax or Bak (Mode 2) (*Llambi et al., 2011*). In the ANTS/DPX liposome dye release assay, BimL-dCTS was functionally comparable to the well-established Bcl-XL inhibitory BH3-protein Bad in reversing Bcl-XL-mediated inhibition of cBid (*Figure 5A*) or Bax (*Figure 5B*). Consistent with the observation that BimL-dCTS was less resistant to displacement by BH3 mimetics in live cells, in cell-free assays BimL-dCTS was also less effective than BimL at displacing cBid or Bax from Bcl-XL (*Liu et al., 2019*). Nevertheless, when assayed with mitochondria BimL-dCTS disrupted the interaction between tBid and Bcl-XL resulting in Bax activation and permeabilization of mitochondria as measured by cytochrome c release (*Figure 5C*, solid black line). This activity is due to inhibition of Bcl-XL function, as in controls without Bcl-XL the same concentration of BimL-dCTS did not directly activate sufficient Bax to mediate MOMP (*Figure 5C*, dashed black line). Thus, purified BimL-dCTS is functional and can initiate MOMP by displacing direct-activators (Mode 1) or activated Bax (Mode 2) from Bcl-XL (*Figure 5A–C*). Finally, BimL-dCTS labeled with Alexa568 retained high-affinity binding for Bcl-XL labeled with Alexa647 both in solution (Kd <16 nM) and on membranes (~35 nM apparent Kd on liposomes and on mitochondria) as measured by FRET (*Figure 6B*).

## Different residues in the Bim CTS regulate membrane binding and Bax activation

To identify which residues in the Bim CTS mediate binding to membranes and/or Bax we generated a series of point mutations. Sequence analysis using HeliQuest software (*Gautier et al., 2008*) predicts that the Bim CTS forms an amphipathic α-helix (*Figure 6A*). Two arginine residues (R130 and 134) are predicted to be on the same hydrophilic side of the helix, whereas hydrophobic residues (e.g. I125, L129, I132) face the other side (*Figure 6A*). To determine the functional importance of these residues, Bim CTS mutants were created including: BimL-CTS2A in which R130 and R134 were mutated to alanine; and a series of single hydrophobic residue substitutions by glutamate (V124E, I125E, L129E, and I132E) (*Figure 6A*). To compare the effects of the CTS mutations on BimL-binding interactions and function, we measured by FRET the Kds for the various binding interactions and the activities of the mutants to promote Bax-mediated liposome permeabilization as EC50's for ANTS release (*Figure 6B* and *Figure 6—figure supplement 1A–E*).

Mutation of individual hydrophobic residues on the hydrophobic side of the Bim CTS (BimL-I125E, BimL-L129E or BimL-I132E) abolished binding to membranes (*Figure 6B*). In contrast, mutations on the other side of the helix including BimL-V124E and BimL-CTS2A had less effect on Bim binding to membranes (*Figure 6B*). Despite the dramatic changes in affinity for membranes among Bim CTS mutants, the mutations did not abolish binding to Bax both in the presence and absence of membranes (*Figure 6B*). Indeed most of the mutants had Kd values for binding to Bax of less than 100 nM and to our surprise many of them bound to Bax better in solution compared to when membranes were present. The presented Kds under 'membranes present' is an estimation of the effective Kd (combined solution and two-dimensional Kds), as we are not able to precisely determine the quantity of protein-complexes that form in solution or on the membrane. Nevertheless, this data reflects the situation in cells and further confirms that binding to membranes and Bax are independent functions of the Bim CTS . In the case of BimL-I125E, a mutant that activates Bax to permeabilize liposomes, the initial interaction with Bax must occur in solution as neither protein spontaneously binds to membranes (*Figure 6B*).

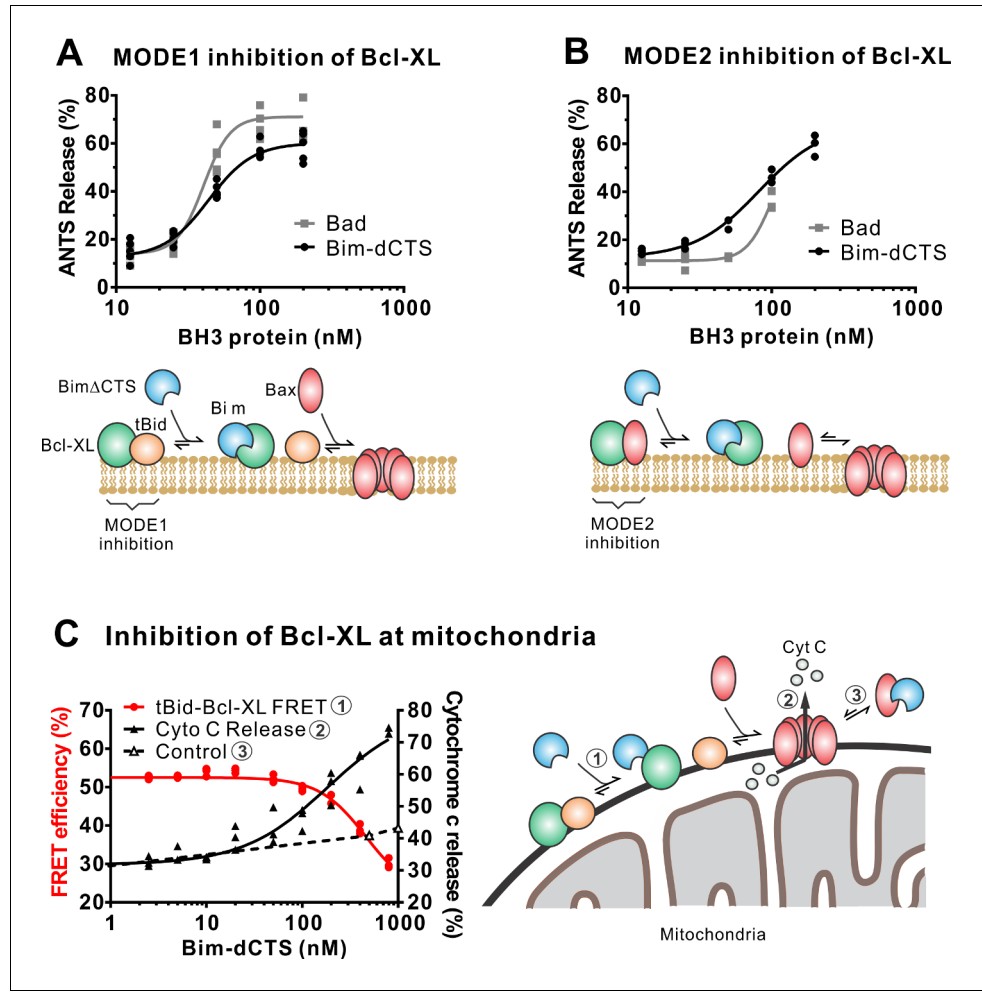

**Figure 5.** The Bim CTS is not required to inhibit Bcl-XL. (A–B) BimL-dCTS and Bad release tBid (A) or Bax (B) from Bcl-XL. 20 nM tBid (A) or tBidmt1 (B) a protein that activates Bax but does not bind Bcl-XL, were incubated with 100 nM Bax, 40 nM Bcl-XL, 0.04 mg/mL ANTS/DPX liposomes, and the indicated amounts of either Bad or BimL-dCTS. Liposome permeabilization was assessed after incubation at 37 °C for 3 hr by measuring the increase in fluorescence due to ANTS/DPX release. Cartoons indicate the interactions measured, BimL (blue), Bax (red), tBid (orange), Bcl-XL (green), and membranes (tan). (C) BimL-dCTS displaced tBid from Bcl-XL and permeabilized mitochondria. Mitochondria were incubated with Bcl-XL (40 nM), tBid (20 nM), Bax (100 nM) and mitochondria. Increasing concentrations of BimL-dCTS were added and displacement of tBid from Bcl-XL was measured by FRET. Mitochondria were then pelleted and cytochrome c release measured by western blotting. Control reactions containing only Bax and BimL-dCTS did not result in cytochrome c release (dotted line). Individual points are shown for three independent replicates. Not all points are visible due to overlap. The adjacent cartoon indicates the interactions measured.

The online version of this article includes the following source data for figure 5:

**Source data 1.** Source data demonstrating that recombinant BimL-dCTS indirectly activates Bax through inhibition of Bcl_XL.

Unexpectedly, there was not a good correlation between BimL binding to membranes and Bax activation. For example, while BimL bound to membranes with a Kd of 31 pM, BimL-CTS2A and BimL-I125E bound to membranes with Kds of ~600 and>1000 pM, respectively yet both mutants triggered Bax-mediated membrane permeabilization, demonstrating that specific residues in the CTS rather than binding to membranes enabled BimL to mediate Bax activation. Moreover, BimL binding to Bax was also not sufficient to activate Bax efficiently. BimL-L129E and BimL-I132E are two Bim mutants that do not bind membranes, retain reasonable affinities for Bax in the presence of membranes (Kds ~ 100–200 nM), but were unable to activate Bax (*Figure 6B*). These results indicate

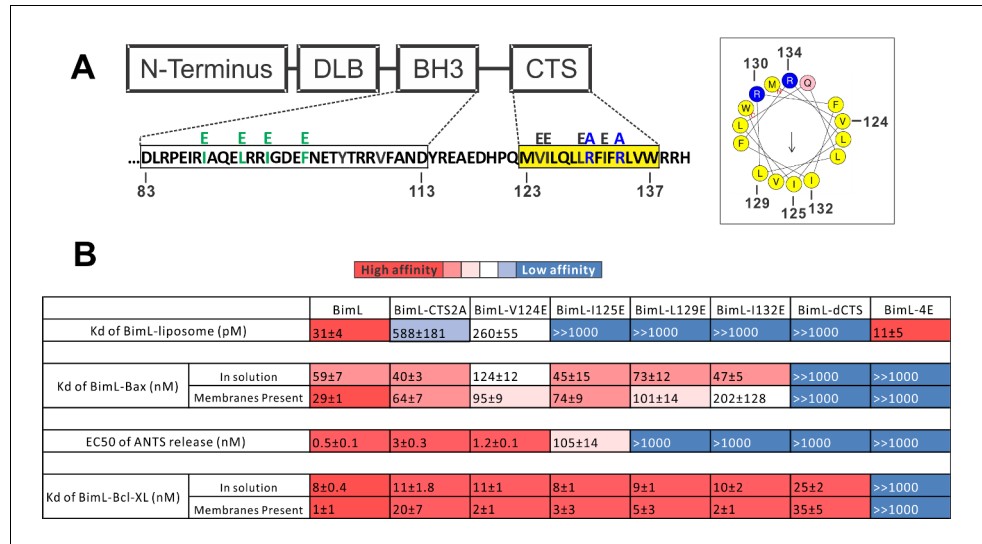

**Figure 6.** Residues within the Bim CTS distinctly regulate membrane binding and Bax activation. (**A**) Diagram of BimL depicting the various domains (DLB: dynein light chain binding motif) and the sequences of the BH3-domain and CTS. The four essential hydrophobic residues in BH3-domain that were mutated to glutamic acid are colored green. Two positive charged residues in the CTS mutated to alanine are colored blue. Glutamic acid mutations for individual hydrophobic residues in the CTS are indicated in black on top of the original sequence. A predicted alpha helix structure generated via HeliQuest software is shown on the right, indicating the amphipathic nature of the CTS. The arrow central to the helix shows the polarity direction for hydrophobicity. The Q indicated in pink is the fifth amino acid in the CTS. Other residues are colored as in the linear sequence. (**B**) Binding of BimL mutants to liposomes, Bax and Bcl-XL expressed as apparent dissociation constants (Kd) measured from raw data as in *Figure 6—figure supplement 1* for each binary interaction. Activation of Bax (EC50) measured from ANTS/DPX assays in *Figure 6—figure supplement 1*. Values are mean ± SEM (n = 3). The table is colour-coded in a heat map fashion as follows: red 0–40; light red 40–80; light pink 80–120; white 120–500; light blue 500–1000; Dark blue >1000. All values are nM except for binding to liposomes which is in pM. The Kds for 'membranes present' measurements are apparent values since diffusion for the protein fraction bound to membranes is in two dimensions while diffusion for the fraction of protein in solution is in three dimensions and several of the binary interactions take place in both locations. Apparent Kd values may also be affected by competing interactions with membranes.

The online version of this article includes the following source data and figure supplement(s) for figure 6:

**Figure supplement 1.** Full titrations using recombinant proteins identifies residues in the Bim CTS important for binding to membranes, and for Bim binding to and activating Bax.

**Figure supplement 1—source data 1.** Source data with fitted curves used to calculate dissociation constants and EC50's for *Figure 6B*.

---

that these two residues play a key function in Bax activation. As expected, the negative control BimL-4E mutant does not bind to nor activate Bax even though its CTS is intact and the protein binds membranes (*Figure 6B*). This result confirms the essential role of the BH3-domain and suggests that the Bim CTS provides a secondary role in Bax binding rather than providing an independent high affinity binding site that is sufficient to activate Bax.

Both functional and binding assays for the various point mutants suggest that specific residues in the Bim CTS participate in Bim-Bax protein interactions that lead to Bax activation; however, these mutants did not clearly separate the membrane binding function of the CTS of Bim from a potential function in Bax activation. Thus, it remains possible that restoring membrane binding to BimL-dCTS would be sufficient to restore Bax activation function. To address this, we fused the mitochondrial tail-anchor from mono-amine oxidase (MAO residues 490–527, UniProt: P21397-1) to the C-terminus of BimL-dCTS to restore membrane binding with a sequence unlikely to contribute to Bax activation directly. This protein, BimL-dCTS-MAO, and BimL bound to mitochondrial-like liposomes and purified mitochondria (*Figure 7A* and *Figure 7—figure supplement 1A* respectively). As expected, a population of these recombinant Bim proteins remains in solution. To directly assess the Bim CTS

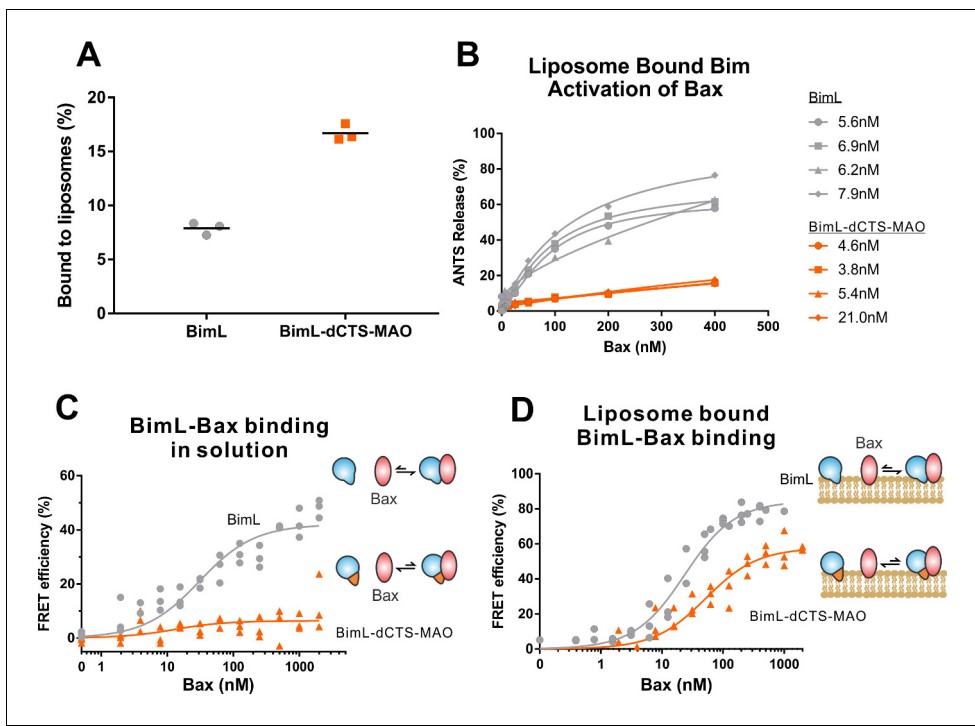

**Figure 7.** BimL-dCTS-MAO binds to liposomes and Bax but activates Bax poorly. Cartoons at the side indicate the measurements being made with equilibria arrows representing the results obtained. Blue objects, Bim. Red ovals, Bax. Blue and orange objects, BimL-dCTS-MAO. (A)BimL-dCTS-MAO targets to liposomes as efficiently as BimL. Alexa568-labeled single cysteine (Q41C) recombinant BimL, and BimL-dCTS-MAO (20 nM) were incubated with 0.5 mg/ml of liposomes for 40 min at 37˚C. Liposomes were then subjected to gel filtration column chromatography, and the Alexa568 fluorescence was measured in the liposome fractions. (B) Restoring membrane binding to BimL-dCTS by adding the MAO tail-anchor sequence (BimL-dCTS-MAO) did not restore Bax activation function. Using gel filtration chromatography, Alexa568-labeled Bim bound to liposomes containing ANTS/DPX were isolated, and the concentration of membrane-bound Bim was calculated based on Alexa568 fluorescence. Increasing concentrations of recombinant Bax protein were then added to these liposomes, and Bax activation was measured as an increase in ANTS release. (C)BimL-dCTS-MAO does not bind to Bax in solution. 10 nM of Alexa568-labeled BimL or BimL-dCTS-MAO was incubated with increasing amounts of Alexa647-labeled Bax. FRET was measured as a decrease in Alexa568 fluorescence signal. (D) Liposome bound labeled BimL-dCTS-MAO binds to Bax. Liposomes were incubated with Alexa568-labeled BimL or BimL-dCTS-MAO as in (B), and incubated with increasing amounts of Alexa647-labeled Bax. FRET was measured as a decrease in Alexa568 fluorescence.

The online version of this article includes the following source data and figure supplement(s) for figure 7:

**Source data 1.** Source data demonstrating that BimL-dCTS-MAO binds to liposomes and Bax, but activates Bax poorly.

**Figure supplement 1.** BimL-dCTS-MAO binds to mitchondrial membranes but binds poorly to Bax.

**Figure supplement 1—source data 1.** Source data demonstrating BimL-dCTS-MAO binds to mitochondria but binds poorly to Bax.

**Figure supplement 2** Recombinant BimL proteins do not aggregate and are the expected moelcular weights.

**Figure supplement 2—source data 1.** FPLC source data demonstrating recombinant BimL proteins do not aggregate.

contribution to the binding and activation of Bax on the membrane surface, we incubated a defined amount of recombinant Alexa568-labeled BimL or BimL-dCTS-MAO proteins with 0.5 mg/mL ANTS/DPX filled liposomes, then isolated the liposomes using size-exclusion chromatography. Using this procedure, we excluded all recombinant Bim that remained in solution, and obtained the membrane-bound BimL or BimL-dCTS-MAO at a concentration of ~5 nM as calculated based on Alexa568 fluorescence intensity (Precise concentrations labeled in *Figure 7B*). This equates to 0.6 Bim molecules per liposome. Addition of Bax to these liposomes resulted in ANTS/DPX release for roughly 60% of the liposomes in incubations containing BimL-membrane-bound liposomes, but no significant

release from the liposomes in incubations containing BimL-dCTS-MAO-bound liposomes (*Figure 7B*), suggesting that the Bim CTS contributes to the activation of Bax even on the membrane surface.

To directly measure binding to Bax, Alexa568-labeled BimL and BimL-dCTS-MAO were incubated with increasing concentrations of Alexa647-labeled Bax protein. In the absence of membranes, no detectable FRET was measured between Alexa568-labeled BimL-dCTS-MAO with Alexa647-labeled Bax (*Figure 7C*), suggesting that the specific sequence of the Bim CTS contributes to the Bim-Bax interaction in solution. However, when Alexa568-labeled BimL-dCTS-MAO bound to the liposome membrane was isolated by size-exclusion chromatography as described above, a FRET signal was detected between it and Alexa647-labeled Bax. We also detected FRET between Alexa568-labeled BimL-dCTS-MAO and Alexa647-labeled Bax in the presence of purified mitochondria (*Figure 7—figure supplement 1B*). Although these incubations contained both soluble and membrane-bound Bim protein, we conclude the interaction occurred on the MOM as no FRET signal was detected between these proteins in solution (*Figure 7C*). Despite measureable binding between BimL-dCTS-MAO and Bax on the liposome membrane (*Figure 7D*), BimL retained a higher affinity than BimL-dCTS-MAO for Bax (Kds of 21 nM and 49.6 nM, respectively). Together these data suggest that the CTS of Bim contributes to both binding to (*Figure 7D*) and activation of Bax (*Figure 7B*) on the liposome membrane.

## Residues within the Bim CTS are proximal to Bax in solution and on mitochondrial membranes

Our binding and mutagenesis data suggest that the Bim CTS binds to and activates Bax in solution and on membranes. To detect this binding interaction, we used a photocrosslinking approach, in which a BimL protein was synthesized with a photoreactive probe attached to a single lysine residue positioned in the CTS using an in vitro translation system containing 5-azido-2-nitrobenzoyl-labled Lys-tRNA$^{Lys}$ that incorporates the lysine analog ($\epsilon$ANB-Lys) into the polypeptide when a lysine codon in the BimL mRNA is encountered by the ribosome. The BimL synthesized in vitro was also labeled by $^{35}$S via methionine residues enabling detection of BimL monomers and photoadducts by phosphor-imaging.

The radioactive, photoreactive BimL protein was incubated with a recombinant His6-tagged Bax protein in the presence of mitochondria isolated from BAK$^{-/-}$ mouse liver lacking endogenous Bax and Bak to prevent competition and increase BimL-Bax protein interactions. Mitochondrial proteins were then separated from the soluble ones by centrifugation. Both soluble and mitochondrial fractions were photolyzed to activate the ANB probe generating a nitrene that can react with atoms in close proximity ($\leq$12 Å from the C$\alpha$ of the lysine residue). Thus, for photoadducts to form, the atoms of the bound Bax molecule are likely to be located in or near the binding site for the Bim CTS. The resulting photoadduct between the BimL and the His$_6$-tagged Bax was enriched by Ni$^{2+}$-chelating agarose resin and separated from the unreacted BimL and Bax monomers using SDS-PAGE. The $^{35}$S-labeled BimL in the photoadduct with His6-tagged Bax and BimL monomer bound to the Ni$^{2+}$-beads specifically via the His$_6$-tagged Bax or nonspecifically were detected by phosphor-imaging. A BimL-Bax-specific photoadduct was detected when the ANB probe was located at four different positions in the Bim CTS on both hydrophobic and hydrophilic surfaces of the potential $\alpha$-helix (*Figure 8A*). These photoadducts have the expected molecular weight for the Bim-Bax dimer, and were not detected or greatly reduced when the ANB probe, the light, or the His6-tagged Bax protein was omitted (*Figure 8A*). Consistent with the BimL-Bax interaction detected by FRET in both solution and membranes, the BimL-Bax photocrosslinking occurred in both soluble and mitochondrial fractions. Less photocrosslinking occurred in the mitochondrial fraction likely due to the fact that in membranes homo-oligomerization of activated Bax competes with hetero-dimerization between BimL and Bax.

As expected, when the ANB probe was positioned in the Bim-BH3 domain as a positive control BimL-Bax photocrosslinking was detected in both soluble and mitochondrial fractions (*Figure 8B*). Crosslinking with the Bim BH3-domain is consistent with the BH3 interaction with the canonical groove or trigger pocket that is well supported by experimental evidence including co-crystal structures and NMR models (*Gavathiotis et al., 2008*; *Robin et al., 2015*). Furthermore, loss of photocrosslinking for BimL mutants with the BH3-4E mutation that abolished binding to Bax demonstrates that direct binding between the proteins is required for crosslinking to be detectable (*Figure 8C*).

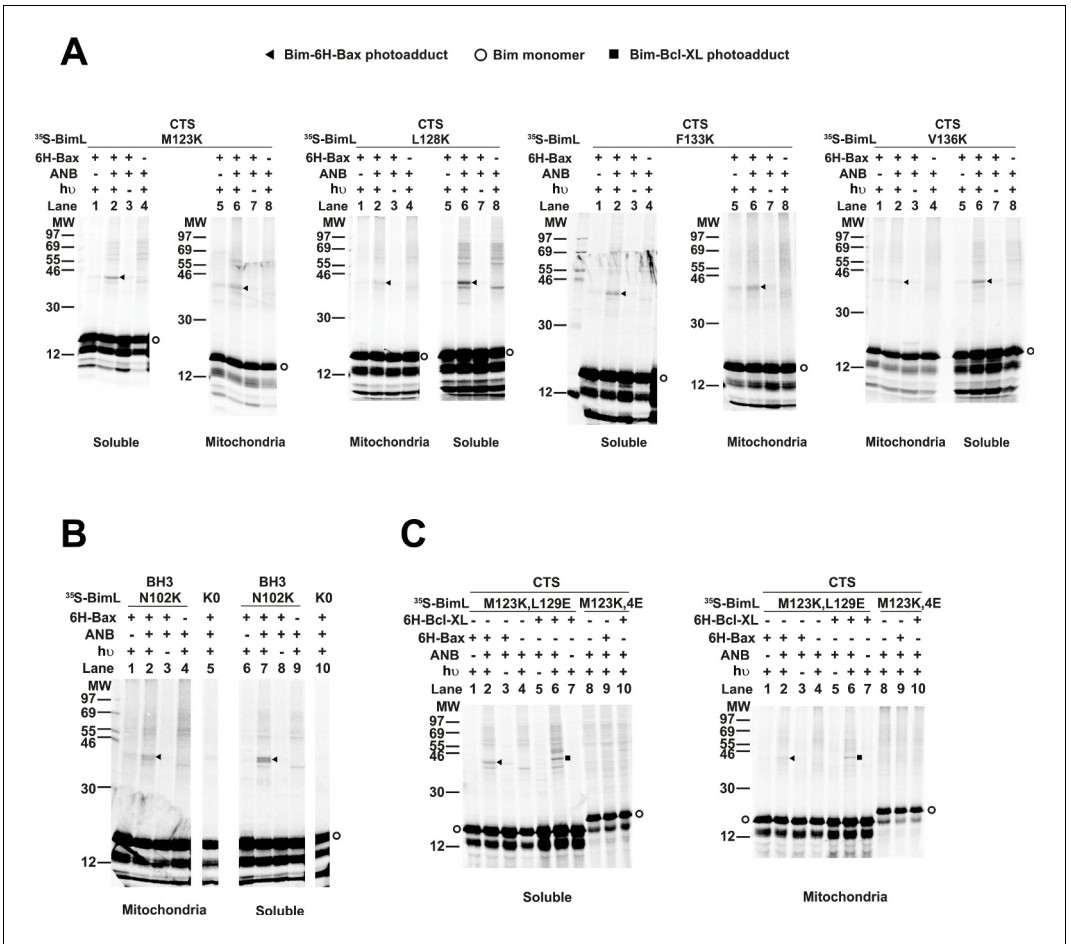

**Figure 8.** Residues within the Bim CTS interact with Bax. (**A**) Interaction of the Bim CTS with Bax in both soluble and mitochondrial fractions detected by photocrosslinking. The [35S]Met-labeled BimL proteins, each with a single εANB-lysine residue located at the position in the CTS indicated at the top of the panels, were synthesized in vitro, and incubated with His6-tagged Bax protein (6H-Bax) in the presence of mitochondria lacking endogenous Bax and Bak. The mitochondria were then separated from the soluble proteins by centrifugation and both fractions were photolyzed. The resulting radioactive BimL/6H-Bax photoadducts were enriched with Ni2+-beads, and analyzed by SDS-PAGE and phosphor-imaging. BimL/6H-Bax dimer specific photoadducts were detected in both mitochondrial and soluble fractions and indicated by arrowheads. They were of reduced intensity or not detected in control incubations in which the ANB probe, light (hυ)or 6H-Bax protein was omitted, as indicated. The radioactive BimL monomers are indicated by circles. The migration positions of protein standards are indicated by molecular weight (MW) in kDa. (**B**) Interaction of the Bim BH3-domain with Bax in both soluble and mitochondrial fractions detected by photocrosslinking. As a positive control experiment illustrating the expected efficiency for the photocrosslinking of BimL protein with a single εANB-lysine located at the indicated position in the BH3-domain was used to photocrosslink 6H-Bax protein. In another control experiment, a lysine-null BimL protein that does not contain any εANB-Lys (**K0**) was used. As expected, a BimL/6H-Bax specific photoadduct was detected in the former but not the latter experiment. And, the efficiencies for the photocrosslinking via photoprobes located in the BH3-domain (**B**) and the CTS (**A**) are comparable strongly suggesting that like BH3-domain, the CTS interacts with Bax directly. (**C**) The 4E mutation in the BH3-domain but not the L129E mutation in the CTS of Bim inhibited photocrosslinking of the Bim CTS to Bax. The BimL protein with either the 4E or the L129E mutation and the εANB-Lys in the CTS was used in the photocrosslinking reaction with either 6H-Bax or 6H-Bcl-XL protein in both soluble and mitochondrial fractions. While the L129E mutation did not inhibit photocrosslinking of BimL to either 6H-protein, the4E mutation did. The BimL/6H-Bcl-XL photoadducts are indicated by squares. The phosphorimages in all panels are representatives from two independent photocrosslinking experiments with each indicated pair of BimL and 6H-Bax or 6H-Bcl-XL proteins.

Therefore, the crosslinking data suggests that similar to the BH3-domain, the Bim CTS binds to Bax. To further demonstrate that the CTS of Bim binds to Bax independent of both membrane binding and Bax activation, the experiment was repeated with BimL-L129E, a mutant that binds Bax without activating it and that does not bind membranes (*Figure 6B*). As shown in *Figure 8C*, the L129E mutant photocrosslinked to Bax in both the soluble and mitochondrial fractions. Furthermore, this mutant also photocrosslinked to Bcl-XL (*Figure 8C*), consistent with data demonstrating that the Bim CTS also binds to this anti-apoptotic protein (*Figure 6B* and *Liu et al., 2019*). The qualitative photo-crosslinking data (*Figure 8*) and the quantitative FRET data (*Figure 6B*) obtained from the BimL proteins with and without the CTS or BH3-domain mutation are consistent, and both support a model in which the CTS interacts with membranes and binds to Bax, thereby enhancing BH3-domain mediated Bax activation.

## The Bim CTS binds to the BH3-binding pocket on Bax

To identify the binding site for the Bim CTS in Bax, we used a chemical crosslinking approach. Unlike the photocrosslinking approach that does not reveal the location of the binding site, the chemical crosslinker bismaleimidohexane (BMH) contains two sulfhydryl reactive moieties separated by a 13 Å spacer, and thus formation of a BMH-crosslinked Bim-Bax dimer requires a cysteine in Bim that is in close proximity with another cysteine in the interacting Bax. Therefore, a successful crosslink indicates a close proximity between the two cysteines, potentially revealing the Bim-binding site in Bax.

We used a structurally well-defined Bim-Bax interaction to validate this crosslinking approach. According to the crystal structure (PDB ID 4ZIE; *Robin et al., 2015*), the BH3-domain of Bim binds to the canonical groove of Bax. Our FRET data shows that BimL and Bax bind in solution and the binding is abolished by the 4E mutation in the Bim BH3-domain that eliminates the nonpolar interactions with the Bax groove. It is therefore expected that this Bim-Bax interaction is mediated by the BH3-domain and the groove, and according to the structure, a BMH molecule would be able to link a cysteine replacing Phe[101] in the BH3-domain of Bim to a cysteine replacing Trp[107] in the canonical groove of Bax. We thus synthesized the [$^{35}$S]Met-labeled single-Cys Bim F101C and Bax W107C proteins in vitro, let them interact in solution, and subjected the sample to BMH crosslinking. When the products were analyzed by SDS-PAGE and phosphor-imaging two BMH-crosslinked products with molecular weights close to that of a BimL-Bax heterodimer were detected (*Figure 9A*, lane 4, indicated by open and closed triangles). The lower molecular weight band indicated by a closed triangle is the BMH-linked BimL-Bax heterodimer since it was absent in the control reactions when either the single-cysteine BimL or Bax was replaced by the respective cysteine-null (C0) protein (*Figure 9A*, lane 6 or 2). The higher molecular weight band indicated by an open triangle is the BMH-linked Bax homodimer since it was also present in the control reaction containing the single-cysteine Bax and the cysteine-null BimL (*Figure 9A*, lane 6). These results demonstrate the BMH crosslinking approach can detect the interaction of the BH3-domain of BimL with the canonical groove of Bax, and hence in principle it can be used to reveal the Bim CTS-binding site in Bax.

Sequence analysis predicts that similar to the BH3-domain the Bim CTS forms an amphipathic α-helix (*Figure 6A*). Sequence alignment revealed a high similarity between the CTS and the BH3-domain as both have the same hydrophobic residues at the h0, h1, h2 and h3 positions, and the same polar or charged residue at the h1+two or h2+one position (*Figure 9D*). The Bim BH3 residues at these positions make critical contacts with the Bax canonical groove that are important for Bax activation (*Robin et al., 2015*; *Weber et al., 2007*). To determine whether the Bim CTS binds to the same Bax groove as the Bim BH3-domain, we performed BMH crosslinking using a Bim W137C mutant that has a single cysteine near the C-terminus of the CTS. Like the Bim F101C mutant with the cysteine near the C-terminus of the BH3-domain, Bim W137C crosslinked to Bax W107C (*Figure 9A*, lane 10, indicated by a closed triangle), suggesting that the BH3-binding groove is also a binding site for the Bim CTS. Consistent with this interpretation, this BimL-Bax heterodimer specific crosslinking did not occur in the control reaction with either single-Cys protein substituted by the respective cysteine-null protein (*Figure 9A*, lane 8 or 12), unlike the Bax homodimer specific crosslinking that also occurred in the control reaction with the single-cysteine Bax and cysteine-null BimL (*Figure 9A*, lane 6). Reciprocal immunoprecipitation by BimL and Bax specific antibodies further identified the crosslinked BimL-Bax heterodimer from the cysteine in the Bim CTS or BH3-domain to the cysteine in the Bax groove (*Figure 9—figure supplement 1*).

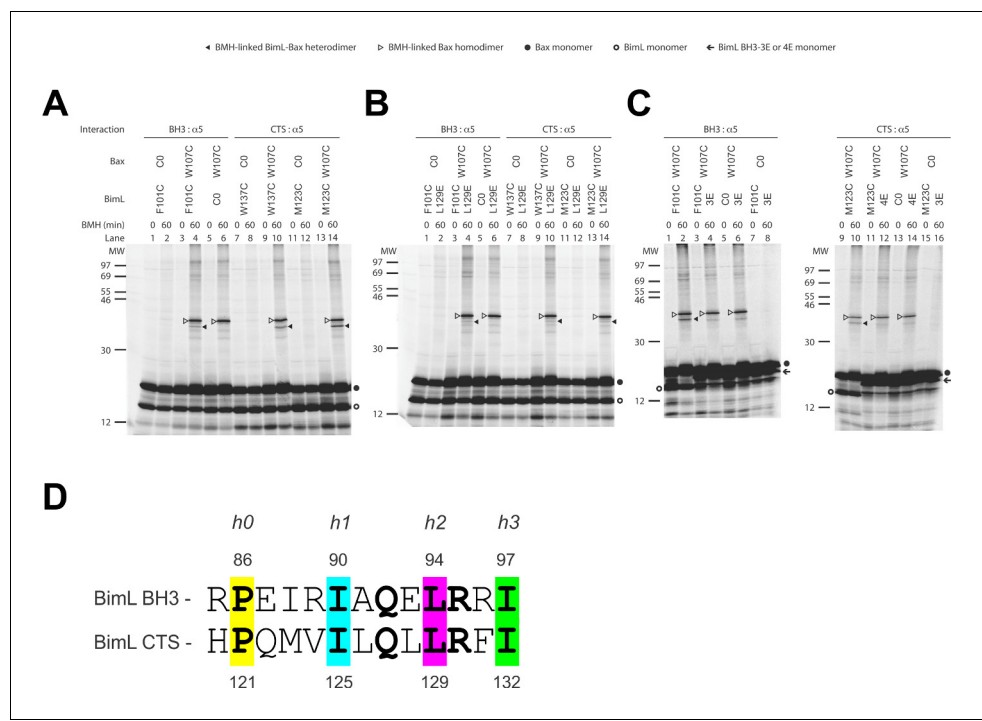

**Figure 9.** The CTS of Bim binds to the canonical BH3-binding groove of Bax. (A) Interaction of the BH3-domain and CTS of Bim with the canonical groove of Bax in solution detected by chemical crosslinking. The radioactive cysteine-null (C0) or single-cysteine BimL and Bax proteins were synthesized in vitro. The indicated BimL-Bax pairs were incubated and subjected to BMH crosslinking for the indicated times. The products were analyzed by SDS-PAGE and phosphorimaging. The BMH-linked BimL-Bax heterodimers (indicated by closed triangles) were detected in the sample containing BimL with F101C in the BH3-domain or M123C or W137C in the CTS and Bax with W107C in the groove. As expected, these BimL-Bax heterodimers were not detected when one of the single-cysteine proteins in each pair was substituted with the respective cysteine-null (C0) protein. In contrast, the BMH-linked Bax homodimers (indicated by open triangles) were detected not only in the samples with each single-cysteine Bax-BimL pair but also that with the single-cysteine Bax and the cysteine-null BimL. The BimL and Bax monomers are indicated by open and closed circles, respectively. The migration positions of protein standards are indicated by MW in kDa. (B–C) The Bim CTS mutation (L129E) reduced, whereas the BH3 mutation (3E or 4E) abolished the interaction of the CTS and BH3-domain with the Bax canonical groove. The BMH crosslinking was done with the indicated single-Cys BimL-Bax pairs to detect the CTS or BH3-groove interaction. As indicated, some BimL proteins contain the L129E or 4E mutation, whereas others contain the 3E (I90E, L94E, and I97E) mutation since residue F101 was changed to Cys instead of Glu. The phosphorimages in panels A to C are representatives from two independent BMH crosslinking experiments with each pair of indicated BimL and Bax proteins. (D)Sequence alignment of the BimL BH3 (residues 85–97) with the BimL CTS (residues 120–132). Note the similarities between the CTS and the BH3-domain as both have the same hydrophobic residues at the h0, h1, h2 and h3 positions, and the same polar or charged residue at the h1+two or h2+one positions.

The online version of this article includes the following figure supplement(s) for figure 9:

**Figure supplement 1.** Identification of the BimL-Bax heterodimer specific BMH crosslinking product by reciprocal immunoprecipitation.

To further define this noncanonical CTS-groove interaction, we repeated the crosslinking using other cysteine positions in the Bim CTS and additional cysteine mutants in the canonical groove in Bax. The only strong heterodimer specific crosslinking detected was between Bim M123C (a cysteine near the N-terminus of the CTS) and Bax W107C (*Figure 9A*, lane 14, indicated by a closed triangle). Since this Bax mutant was also crosslinked to the C-terminus of CTS via W137C, binding of the CTS to the groove seems to occur in either orientation. Additional weak but specific BimL to Bax crosslinking was detected from M123C to E69C and W137C to D98C providing further support for this flexible interaction (*Figure 9—figure supplement 1*).

To determine whether the physical interaction between the Bim CTS or BH3-domain and the Bax groove detected by the crosslinking is functional, we tested the effect of the L129E mutation in the Bim CTS or the 4E mutation in the Bim BH3-domain on the crosslinking because both mutations greatly inhibited the activation of Bax by BimL (*Figure 6B*). We found that the L129E mutation reduced the Bim CTS to Bax interaction with the binding groove detected by the Bim W137C or W123C to Bax W107C crosslinking (compare the closed triangle-indicated band in *Figure 9B*, lane 10 or 14 with that in *Figure 9A*, lane 10 or 14). Since we used Bim F101C to detect the BH3-domain interaction with the Bax groove, and the F101 was changed to E in the 4E mutant, we generated Bim F101C/3E mutant to assess the effect of the BH3 mutation. As expected, the 3E mutation abolished Bim F101C crosslinking to Bax W107C, and hence, the BH3-interaction with the groove (*Figure 9C*, the closed triangle indicates the band in lane two that disappeared in lane 4 when the 3E mutant was used). Surprisingly, mutation of the CTS also inhibited the BH3-interaction with the groove (*Figure 9B*, lane four vs. *Figure 9A*, lane 4), while the BH3 mutation also abolished the CTS interaction with the groove (*Figure 9C*, lane 12 vs. lane 10), suggesting that the two interactions are not independent. The BH3-domain may contribute more than the CTS to the overall protein-protein interaction based on the severity of the effect of mutations on the crosslinking and FRET (*Figure 6B*). Together, the crosslinking data from these loss-of-function Bim mutants demonstrate that the CTS and BH3-interactions with the groove detected between the soluble Bim and Bax proteins are functionally important for Bim mediated activation of Bax.

## Bim CTS mutants that cannot activate Bax in vitro do not kill HEK293 cells

Together, our data suggests that specific residues within the Bim CTS are involved in different aspects of BimL function. Residue I125 is required for Bim to bind to mitochondria but is of lesser importance in activating Bax. In contrast, residues L129 and I132 are not required for BimL to bind Bax but are important for it to efficiently activate Bax. Finally, BimL-dCTS functions only to bind and inhibit Bcl-XL. The defined mechanism(s) of these mutants makes them useful for probing the differential sensitivity of HEK293 and MEF cells to expression of VBimL-dCTS as seen in *Figure 1*. Expression of the mutants in HEK293 cells by transient transfection revealed that similar to VBimL-dCTS, expression of either VBimL-L129E or VBimL-I132E was not sufficient to kill HEK293 cells, despite expression of either mutant being sufficient to kill the primed MEF cell line (*Figure 10*). In contrast, HEK293 cells were killed by expression of VBimL-I125E, albeit to a lesser extent than by VBimL (*Figure 10*). This result is consistent with our findings with purified proteins showing that the EC50 for liposome permeabilization by BimL-I125E was 100 nM compared to ~1 nM for BimL (*Figure 6B*). The activity of VBimL-I125E also demonstrates that BimL binding to membranes is not required to kill HEK293 cells as BimL-I125E does not bind membranes (*Figure 6B*). Together, this data suggests that unlike MEF cells, only mutants of BimL that can efficiently activate Bax kill HEK293 cells.

## Discussion

The apoptotic activity of Bim in live cells is likely mediated by a combination of functions that result in both activation of Bax and inhibition of anti-apoptotic proteins. Unlike any of the known BH3-proteins or small molecule inhibitors, BimL-dCTS inhibits all of the major multi-domain Bcl-2 family anti-apoptotic proteins without activating Bax or Bak. Thus expression of this tool protein in cells enables new insight into the importance of the extent to which a cell depends on the expression of anti-apoptotic proteins for survival (*Figure 1A*, *Figure 2B*). Our results strongly suggest that the varying levels of apoptotic response of cell lines to BimL-dCTS reflect the extent to which that particular cell type is primed. Thus, HEK293 cells and mature neurons that are resistant to inhibition of Bcl-2, Bcl-XL and Mcl-1 but sensitive to activation of Bax, are functionally unprimed. Partial resistance to expression of BimL-dCTS suggests that the flow of Bcl-2 family proteins between different binding partners leads to differential levels of dependency on the activation of Bax to trigger apoptosis. To illustrate this, we have created a schema illustrating protein flow at the two extremes represented by fully unprimed and primed cells and the effects of mutations in the Bim CTS on regulating apoptosis (*Figure 11*). In the schema, flow is indicated by the different lengths of the equilibria arrows and illustrates the consequences of the various dissociation constants displayed in *Figure 6B*. While BimL efficiently recruits Bax to membranes and activates it, BimL-I125E does not bind to membranes

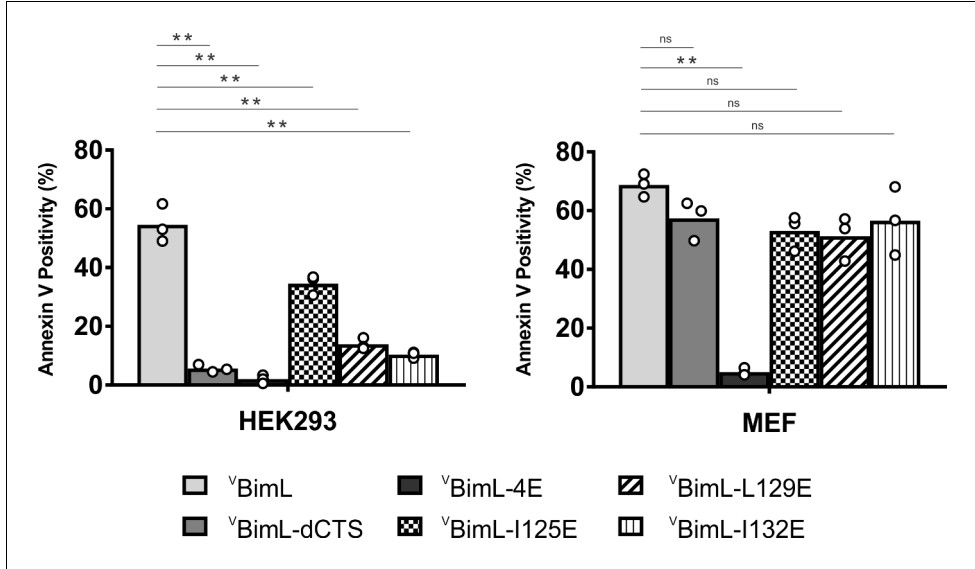

**Figure 10.** Bim CTS mutants that cannot activate Bax in vitro cannot kill HEK293 cells. The indicated cell lines were transiently transfected with DNA to express $^V$BimL, and the indicated $^V$BimL mutant proteins. Cells expressing Venus fusion proteins were stained with the nuclear dye Draq5 and rhodamine-labeled Annexin V, and apoptosis was assessed by confocal microscopy as in *Figure 1*. The y-axis indicates Annexin V Positivity (%), which was calculated based on the total number of Venus expressing cells that also score positive for Annexin V rhodamine fluorescence. A minimum of 400 cells were imaged for each condition. Individual points (open circles) represent the average for each replicate, while the bar heights, relative to the y-axis, represent the average for all three replicates. A one-way ANOVA was used within each cell line followed by a Tukey's multiple comparisons test to compare the means of each transfection group. *p-values<0.05, **p-values<0.01, ns, non-significant p-values (>0.05).

The online version of this article includes the following source data for figure 10:

**Source data 1.** Source data demonstrating that mutants of BimL that cannot activate Bax do not kill H.EK293 cells.

and has reduced binding to Bax therefore this mutant activates Bax less efficiently than BimL (*Figure 11A*). BH3-proteins that do not efficiently activate Bax, such as BimL-L129E or BimL-I132E, do not result in MOMP in unprimed cells (*Figure 11A*) instead they interact primarily with anti-apoptotic proteins (illustrated here as Bcl-XL since it was possible to measure binding with this purified protein). The binding measurements in *Figure 6B* allow prediction of the outcome of more subtle differences in interactions for BimL and its mutants. For example, even though BimL-I125E activates Bax the concentration required is around 100 nM while the dissociation constant for Bcl-XL is less than 3 nM (*Figure 6B*) such that in cells BimL-I125E would preferentially bind and inhibit Bcl-XL rather than activate Bax (*Figure 11B*). While the CTS is necessary for Bim to activate Bax at physiologically relevant concentrations, membrane binding mediated by the CTS is not a prerequisite for interaction with Bax. Rather, binding to membranes increases subsequent Bax activation possibly through facilitating Bax conformational changes on the membrane (*Figure 6B* compare BimL, BimL-CTS2A and BimL-I125E). Thus, it is likely that in cells expressing endogenous Bim, binding to membranes contributes to the efficiency with which the protein kills cells. In addition, a recent publication suggests that membrane-bound Bim is more pro-apoptotic as it can dimerize using the LC1-motif by binding to DLC1 (*Singh et al., 2017*). Here, we observed that one reason that BimL-dCTS is less pro-apoptotic is loss of binding to membranes where it can activate Bax more efficiently. Future work will determine the relative importance of the CTS-mediated Bim binding to mitochondrial membranes and direct activation of Bax. In addition, it will be critical to determine the role for the DLC1-mediated Bim complex formation on binding anti-apoptotic proteins in the membranes. A key question is whether the resulting alterations of Bim interactions with other Bcl-2 family proteins that are mediated by BH3-domain, the CTS and LC1 region are regulated differently in different cell types. Nevertheless, restoring membrane binding to BimL-dCTS using a mitochondrial tail-anchor (BimL-dCTS-MAO) did not restore Bax activation, thus indicating that the Bim CTS does more than

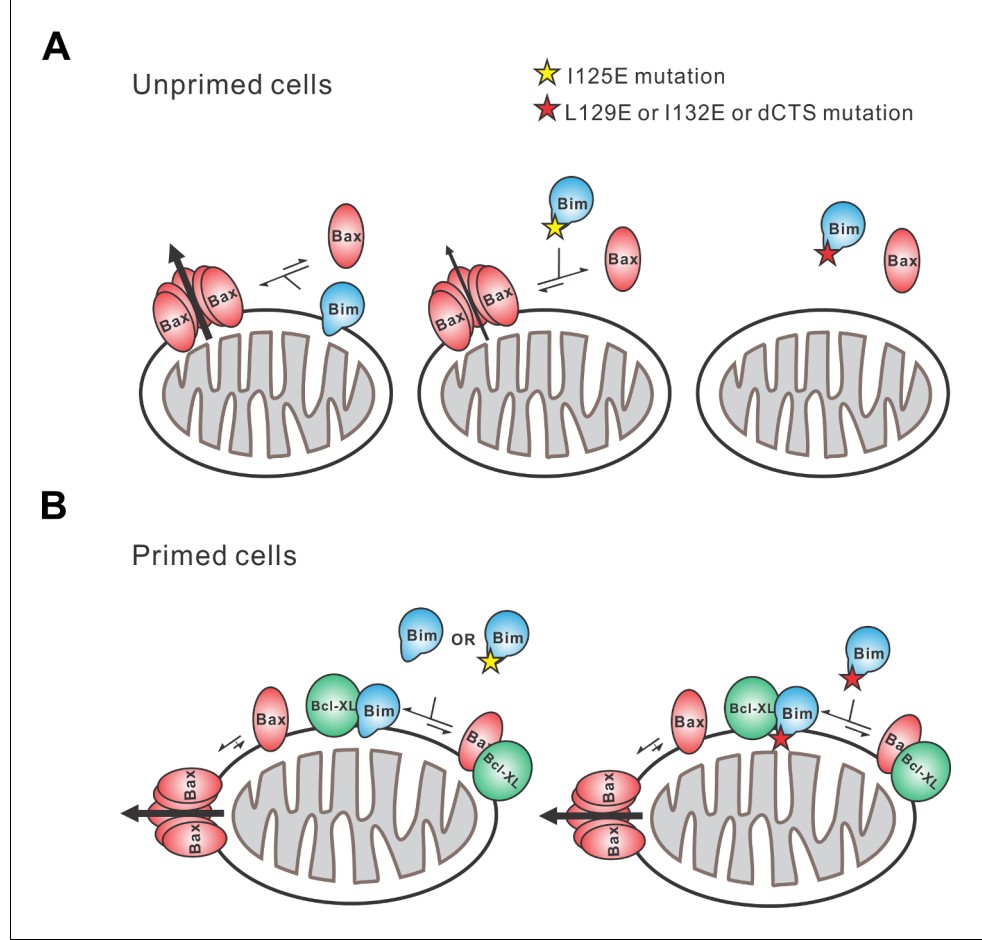

**Figure 11.** Schematic overview for the Bim CTS pro-apoptotic function Interactions between BimL (blue), effector protein Bax (red) and Bcl-XL (green) are shown at mitochondria of unprimed and primed cells as indicated. Mutations of the Bim CTS are shown as a red or yellow star. Direction of protein flow into complexes indicated by lengths of the equilibria arrows is based on the Kds measured for the binding interactions (*Figure 6B*), using the approximate cellular concentrations of the various proteins and activity assays with liposomes and mitochondria. (A) In unprimed cells the direct activation of Bax is the main function of Bim for inducing apoptosis. Comparison of BimL-I125E with BimL-L129E, BimL-I132E, and BimL-dCTS shows that the CTS, not membrane binding controls the activation of Bax by BimL (*Figure 6B*). BimL binds membranes and can activate Bax. BimL-I125E (yellow star) has no detectable membrane-binding activity but still binds to and activates Bax, albeit with reduced activity compared to BimL (*Figure 6B*). At physiologically relevant concentrations BimL-L129E, BimL-I132E, and BimL-dCTS do not activate Bax and do not bind to membranes. However unlike BimL-dCTS. BimL-L129E and BimL-I132E binding to Bax is not reduced enough to account for the loss in Bax activation and membrane permeabilization suggesting these two residues are involved in activating Bax. (B) In primed cells, one or more pro-apoptotic proteins (activated Bax/Bak and/or a Bax/Bak-activating BH3-protein) are sequestered by anti-apoptotic proteins at the MOM. For simplicity only active Bax is shown. Depending on the amount of active pro-apoptotic protein sequestered and the amount of free inactive Bax and or Bak in the cell, BimL may initiate apoptosis primarily by inhibiting anti-apoptotic proteins or by activating Bax and inhibiting anti-apoptotic proteins. The Bim CTS is not required for binding to and inhibiting anti-apoptotic proteins as BimL-L129E, BimL-I132E, and BimL-dCTS bind to anti-apoptotic proteins such as Bcl-XL and release both pro-apoptotic BH3-proteins and Bax (*Figure 5* and *Figure 6B*), thus enabling killing of primed cells.

bind the protein to membranes . Additionally, our work suggests a newly characterized mechanistic distinction between Bim and tBid, as tBid requires membrane binding and a subsequent conformational change in order to bind and efficiently activate Bax (*Lovell et al., 2008*; *Shamas-Din et al., 2013a*) while BimL can activate Bax in solution via dual interactions with the Bim BH3 domain and CTS (*Figures 3C*, *8*, *9* and *11*).

The activities of the various Bim mutants analyzed here further suggest that specific residues in the Bim CTS enable physiological concentrations of Bim to activate Bax. That BimL-V124E, BimL-I125E and BimL-L129E all bind Bax in solution and in the presence of membranes with similar affinities yet vary functionally to trigger Bax-mediated liposome permeabilization by three orders of magnitude, suggests a specific role for this region in activation of Bax (*Figure 6B*) rather than the region simply increasing overall binding affinity. The situation is further complicated by another major role of the CTS of Bim in binding the protein to membranes. BimL-dCTS-MAO binds to mitochondria and liposomes yet is defective in activating Bax to permeabilize these membranes further suggesting a role for specific residues in the CTS binding to and activating Bax (*Figure 7B*). Such a role is consistent with our crosslinking data suggesting direct binding between the CTS of Bim and Bax (*Figure 8*) that is surprisingly mediated at least in part by the canonical BH3-binding groove in Bax (*Figure 9*). Furthermore, these CTS residues particularly L129 (which corresponds to L185 in BimEL) increased the affinity for Bim binding to Bcl-XL such that it conferred resistance to BH3 mimetic drugs (*Liu et al., 2019*). Nevertheless, it remains formally possible that changes in binding affinity coupled with alterations in effective off-rate due to membrane binding may also contribute to the activation of Bax by Bim. In addition to loss of interactions with the membrane BimL-dCTS still retains a greater propensity to activate Bax in comparison to Bim BH3-peptides (Compare *Figure 3A* with *Sarosiek et al., 2013*; Figure 4H). While 1 µM of BimL-dCTS was sufficient to activate Bax in our liposome release assay, over 10 µM of Bim BH3-peptide was required to achieve similar Bax activation. This result suggests that in addition to the BH3 domain and the CTS, other regions of Bim contribute to Bim's pro-apoptotic function as previously suggested (*Singh et al., 2017*). Moreover, it is unclear to what extent other regions of Bim contribute to the BH3 region and the CTS adopting an optimal structure that can efficiently activate Bax. Similar to Bim, full length Bid cleaved by caspase is nearly 10-fold more active than a Bid-BH3 peptide in activating Bax, suggesting other regions outside the BH3-domain of Bid contribute to the interaction and/or activation of Bax. Thus, future studies of Bim and Bid should address what additional sequences other than the BH3-domains and C-terminal regions participate in Bax activation.

Currently, BH3-profiling is the technique used to assay the state of apoptotic priming for different tissue types, however, this technique requires the addition of BH3-peptides at high concentrations, and can only be performed on cells/tissues after permeabilization of the plasma membrane (*Potter and Letai, 2016*). As an alternative, we propose lentiviral delivery and expression of BimL-dCTS be performed on living cells (or tissue samples), with readouts currently being used to assay cell death such as Annexin V staining, condensed nuclei, PI staining of nuclei, etc. Recently, it was reported in adults that most tissues are unprimed (*Sarosiek et al., 2017*); however, the status of priming for different cell types that make up a single tissue may differ. In contrast, in tissue culture most cells are at least partially primed (*Figure 1*). We speculate that stress responses that result when fully or partially transformed cell lines are grown under non-physiological conditions (high glucose and oxygen, in the presence of serum and on plastic with abnormal stromal interactions) generally account for the dependence of these cell lines on continued expression of anti-apoptotic proteins. BH3-profiling can only provide an answer at the tissue level or for cell populations that can be isolated in sufficient quantities or easily cultured (*Sarosiek et al., 2017*). However, lentiviral delivery and expression of ᵛBimL-dCTS in cells in co- or organoid-cultures, tissue slices and in vivo can provide a means to assay the level of dependence of individual cells on expression of anti-apoptotic Bcl-2 family proteins. This information could prove valuable to understanding which cell types may be most affected by small molecule BH3 mimetics as chemotherapeutics and for other drugs to better predict and prevent off-target toxicities that result in cell priming.

Overall, our data suggests a model in which the unusual CTS of Bim is not only required for binding to membranes but is directly involved in the activation of Bax. This interaction likely occurs via binding of the Bim CTS to the BH3-binding groove on Bax (*Figure 9*). However, the BH3-binding groove may not be the exclusive binding site for the Bim CTS. As seen in *Figure 8C*, the CTS of Bim photocrosslinks to 6H-Bax even in the presence of the L129E mutation. However, crosslinking is abolished though with the BH3 4E mutation. How Bim binding to the BH3 binding groove by both its BH3 region and CTS enables Bax activation is the subject of ongoing studies. Nevertheless, this function is crucial for BimL to kill unprimed cells. The CTS also increases the affinity of Bim for binding to Bcl-XL and Bcl-2 (*Figure 11*). The very much higher affinity of Bim for Bcl-XL and Bcl-2 compared to Bax ensures that in cells with excess anti-apoptotic proteins Bim is effectively sequestered

and neutralized. In previous studies, we demonstrated that the additional affinity of the interaction of Bim with Bcl-XL and Bcl-2 provided by the Bim CTS is sufficient to dramatically reduce displacement of Bim by small molecule BH3 mimetics (*Liu et al., 2019*). Thus, regulation of apoptosis by Bcl-2 proteins is more complicated than presented in most current models. Moreover, the mutants and binding affinities described here provide the tools necessary for future studies of the relative importance of activation of Bax compared to inhibition of anti-apoptotic proteins in intact cells and in animals.

# Materials and methods

## Key resources table

| Reagent type (species) or resource | Designation | Source or reference | Identifiers | Additional information |
|---|---|---|---|---|
| Antibody | antibody to Cytochrome c | In house (*Billen et al., 2008*) | | (1:2000) Dilution |
| Antibody | Donkey anti-rabbit (polyclonal) | Jackson Immuno Research Laboratories | Cat. #: 711-035-150 | (1:10000) Dilution |
| Antibody | Donkey anti-mouse (polyclonal) | Jackson Immuno Research Laboratories | Cat. #: 711-035-152 | (1:10000) Dilution |
| Antibody | Antibody to Bax | In house (*Zhu et al., 1996*) | Max6 | (1:1000) dilution |
| Antibody | Antibody to Bim | Santa Cruz Biotechnology | Cat. #: sc-11425 | (1:50) dilution |
| Cell line (*M. musculus*) | Baby Mouse Kidney (BMK)-DKO (Bax and Bak knockout) cells | Other (*Degenhardt et al., 2002*) | | Provided by Dr. Eileen White (Rutgers University) Mycoplasma free, see Materials and methods |
| Cell line (*H. sapiens*) | Cama-1 | | RRID: CVCL_1115 | Provided by Dr. Linda Penn (University of Toronto). Mycoplasma Free, see Materials and methods |
| Cell line (*H. sapiens*) | HEK293 | Other (*Graham et al., 1977*) | RRID: CVCL_0045 | Provided by Dr. Frank Graham (McMaster University). Mycoplasma Free , see Materials and methods |
| Cell line (*H. sapiens*) | HCT-116 | Other (*Polyak et al., 1996*) | RRID: CVCL_0291 | Provided by Dr. Bert Vogelstein (John Hopkins University). Mycoplasma Free , see Materials and methods |
| Strain (*M. musculus*) | Embryonic day 15 embryos | The Jackson Laboratory | C57BL/6J | Used for the preparation of primary cortical neurons and for purification of mitochondria, see Materials and methods. |
| Cell line (*M. musculus*) | MEF | Other (*Pagliari et al., 2005*) | RRID: CVCL_U630 | Provided by Dr. Doug Green (St. Judes Children's Research Hospital) Mycoplasma Free, see Materials and methods |
| Chemical compound, drug | Draq5 | ThermoFisher Scientific, Molecular probes | Cat. #62251 | Nuclear stain for live cell imaging |
| Chemical compound, drug | Hoescht 33258 | Cell signaling technologies | Cat. # 4082S | Nuclear stain for live cell imaging |
| Chemical compound, drug | Tetramethylrhodamine, Ethyl Ester, Perchlorate (TMRE) | ThermoFisher Scientific, Molecular probes | Cat. # T669 | Used to stain actively respiring mitochondria |

*Continued on next page*

*Continued*

| Reagent type (species) or resource | Designation | Source or reference | Identifiers | Additional information |
|---|---|---|---|---|
| Chemical compound, drug | Propidium iodide | Bioshop | Cat. # PPO888.10 | Nuclear stain for dead cells |
| Chemical compound, drug | Alexa 647-maleimide | ThermoFisher Scientific, Molecular probes | Cat. #: A20347 | Acceptor fluorophore in FRET experiments, when Alexa 568 is the donor. |
| Chemical compound, drug | Alexa568-maleimide | ThermoFisher Scientific, Molecular probes | Cat. #. A20341 | Donor fluorophore in FRET experiments when Alexa 647 is the acceptor. |
| Chemical compound, drug | ANTS (8-Aminonaphthalene-1,3,6-Trisulfonic Acid, Disodium Salt) | ThermoFisher Scientific, Molecular probes | A350 | Fluorophore used in liposome release assay (*Billen et al., 2008*) |
| Chemical compound, drug | DPX (p-Xylene-Bis-Pyridinium Bromide) | ThermoFisher Scientific, Molecular probes | X1525 | Quencher used in liposome release assay (*Billen et al., 2008*) |
| Chemical compound, drug | IANBD Amide (N,N'-Dimethyl-N-(Iodoacetyl)-N'-(7-Nitrobenz-2-Oxa-1,3-Diazol-4-yl) Ethylenediamine) | Molecular Probes | Cat. #: D-2004 | Chemically reactive environment sensitive fluorophore. Reacts with Cysteine used to label proteins for in vitro study. |
| Chemical compound, drug | PC (L-α-phosphatidylcholine) | Avanti Polar Lipids | Cat. #: 840051C | For making liposomes, used 48% PC |
| Chemical compound, drug | DOPS (1,2-dioleoyl-sn-glycero-3-phospho-L-serine) | Avanti Polar Lipids | Cat. #: 840035C | For making liposomes, used 10% DOPS |
| Chemical compound, drug | PI (L-α-phosphatidylinositol) | Avanti Polar Lipids | Cat. #: 840042C | For making liposomes, used 10% PI |
| Chemical compound, drug | PE (L-α-phosphatidy lethanolamine) | Avanti Polar Lipids | Cat. #: 841118C | For making liposomes, used 28% PE |
| Chemical compound, drug | TOCL, (18:1 Cardiolipin) | Avanti Polar Lipids | Cat. #: 710335C | For making liposomes, used 4% TOCL |
| Chemical compound, drug | bismaleimidohexane (BMH) | Pierce | 22330 | Chemical crosslinker, Cysteine specific. Used for chemical crosslinking of Bim and Bax proteins, see Materials and methods. |
| Commercial assay or kit | Fugene HD | Promega | Cat. #: E2311 | Transfection reagent for mammalian cells |
| Commercial assay or kit | TransIT-X2 | Mirus | Cat. #: Mir 6003 | Transfection reagent for mammalian cells |
| Gene (*H. sapiens*) | Bax | In house (*Yethon et al., 2003*) | GI: L22473.1 | Expression plasmid for production of recombinant protein |
| Recombinant DNA reagent (*H. sapiens*) | Bax | In house (*Zhang et al., 2010*; *Zhang et al., 2016*) | GI: L22473.1 | For recombinant 6H-Bax protein used in photocrosslinking and for making Cys-null or single Cys recombinant protein in chemical crosslinking |
| Recombinant DNA reagent (*H. Sapiens*) | Bcl-XL | In house (*Ding et al., 2014*) | GI: Z23115.1 | For recombinant 6H-Bcl-XL protein used in photocrosslinking, membrane permeabilization, and protein-protein binding assays. |
| Gene (*H. sapiens*) | Bad | In house (*Aranovich et al., 2012*) | GI: AB451254.1 | For expression of [V]Bad in cells |
| Gene (*M. musculus*) | Bid | In house (*Lovell et al., 2008*) | GI: NM_007544.4 | For recombinant cBid purification ( |
| Gene (*M. musculus*) | BimL | This paper | GI: AAD26594.1 | This lab, plasmid # 2187, for recombinant BimL purificaton |

*Continued on next page*

*Continued*

| Reagent type (species) or resource | Designation | Source or reference | Identifiers | Additional information |
|---|---|---|---|---|
| Recombinant DNA reagent (*M. musculus*) | BimL | This paper | GI: AAD26594.1 | Dr. Lin lab, plasmidpSPUTK-BimL For the single-Cys proteins used in photo and chemical crosslinking |
| Recombinant DNA reagent (*M. musculus*) | tBid | In house (*Aranovich et al., 2012*) | GI: NM_007544.4 | For expression of $^V$tBid in cells |
| Other | Cell Carrier-384, Ultra plate | PerkinElmer | Cat. #: 6057300 | For mono-layer culturing and imaging cell lines |
| Other | Greiner Bio-one Cell culture microplate, 384 well | Greiner Bio-one | Cat. 781090 | For culturing and imaging primary cortical neurons. |
| Other | Non-binding surface, 96-well plate, black with clear bottom | Corning | Cat. #: 3881 | For recombinant protein and liposome assays. It is critical to use a non-binding plate. |
| Other | Opera Phenix | PerkinElmer | Cat. #: HH14000000 | Automated confocal microscope. Used for imaging cell lines and primary cortical neurons. |
| Other | AnnexinV*Alexa647 | In House (*Logue et al., 2009*) | | Used for detecting phosphotidylserine externalization (*Blankenberg et al., 1998*) |
| Other | εANB-[14C]Lys-tRNALys | tRNA Probes | L-32 | Used for incorporation εANB-Lys into Bim protein using an in vitro translation system. The εANB-group is photoactive and generates a nitrene for photocrosslinking |
| Other | [$^{35}$S]Methionine | PerkinElmer | NEG009C | Used for incorporation [$^{35}$S]Met into Bim and Bax proteins using an in vitro translation system for photo or chemical crosslinking, see Materials and methods. |
| Other | transcription/translation (TNT)-SP6 coupled wheat germ extract system | Promega | L4130 | Used for synthesis of Bim and Bax proteins for chemical crosslinking |
| Other | multi-purpose image scanner | Fuji Film | FLA-9000 | Used for phosphorimaging to detect radioactive proteins in gels |
| Software, algorithm | GraphPad Prism | San Diego, California | Version 6 RRID:SCR_002798 | Scientific graphing program, used for curve fitting of in vitro data and to perform statistical analysis. |
| Software, algorithm | ImageJ | PMID: 17936939 | MBF - ImageJ for microscopy, Dr. Tony Collins (McMaster University) | For band density measurements used to quantify Cytochrome c release from immunoblots. |
| Software, algorithm | Multi Gauge | Fuji Film | Version 3.0 | Used for processing and displaying phosphor-images |
| Recombinant DNA reagent (*A. victoria*) | mVenus-pEGFP-C1 | Other | GI: KU341334.1 | Dr. Ray Truant (McMaster University). Backbone EGFP-C1 (Clonetech) |

## Protein purification

Wild type and single cysteine mutants of Bax, Bcl-XL, and cBid were purified as described previously (*Kale et al., 2014*). cBid mutant 1 (cBidmt1) was purified with the same protocol used for cBid (*Kale et al., 2014*). Bad was purified as described previously (*Lovell et al., 2008*). His-tagged Bax and Bcl-XL proteins were purified as described previously (*Ding et al., 2014*)

His-tagged Noxa was expressed in *E. coli* strain BL21DE3 (Life Tech, Carlsbad, CA). *E. coli* cells were lysed by mechanical disruption with a French press. The cell lysate was diluted in lysis buffer (10 mM HEPES (7.2), 500 mM NaCl, 5 mM MgCl2, 0.5% CHAPS, 1 mM DTT, 5% glycerol, 20 mM Imidazole) and Noxa was purified by affinity chromatography on a Nickel-NTA column (Qiagen, Valencia, CA). Noxa was eluted with a buffer containing 10 mM HEPES (7.2), 300 mM NaCl, 0.3% CHAPS, 20% glycerol, 100 mM imidazole, dialyzed against 10 mM HEPES 7.2, 300 mM NaCl, 10% glycerol, flash-frozen and stored at −80℃.

Purification of BimL and single cysteine mutants of BimL was carried out as previously described (*Liu et al., 2019*). Briefly, cDNA encoding full-length wild-type murine BimL was introduced into pBluescript II KS(+) vector (Stratagene, Santa Clara, CA). Sequences encoding a polyhistidine tag followed by a TEV protease recognition site (MHHHHHHGGSGGTGGSENLYFQGT) were added to create an in frame fusion to the N-terminus of BimL. All the purified BimL proteins used here retained this tag at the amino-terminus. However, control experiments demonstrated equivalent activity of the proteins before and after cleavage with TEV protease (Data not shown). Mutations as specified in the text were introduced into this sequence using site-directed mutagenesis.

BimL was expressed in Arabinose Induced (AI) *E. coli* strain (Life Tech, Carlsbad, CA). *E. coli* were lysed by mechanical disruption with a French press. Proteins were purified from the cell lysate by affinity chromatography using a Nickel-NTA column (Qiagen, Valencia, CA), and eluted with a solution containing 20 mM HEPES pH7.2, 10 mM NaCl, 0.3% CHAPS, 300 mM imidazole, 20% Glycerol. The eluate was adjusted to 150 mM NaCl and applied to a High Performance Phenyl Sepharose (HPPS) column. Bim was eluted with a no salt buffer and dialyzed against 10 mM HEPES pH7.0, 20% glycerol, flash-frozen and stored at −80℃.

## Protein labeling

Single cysteine mutants of Bax, Bcl-XL, cBid and Bad were labeled with the indicated maleimide-linked fluorescent dyes as described previously (*Pogmore et al., 2016*; *Kale et al., 2014*; *Lovell et al., 2008*). Single cysteine mutants of Bim were labeled with the same protocol as cBid with the exception that the labeling buffer also contained 4M urea.

## Bim binding to membranes

Liposomes (100 nm diameter) with a lipid composition resembling MOM were prepared as described previously (*Kale et al., 2014*). Mouse liver mitochondria were isolated from Bak-/-mice as previously described (*Pogmore et al., 2016*). Liposomes and mitochondria were labeled with 0.5% and 2% mass ratios of DiD, respectively (Life Tech, Carlsbad, CA). The single-cysteine mutant of BimL, BimL Q41C, was labeled with Alexa568-maleimide and incubated with the indicated amount of unlabeled or DiD-labeled mitochondria or liposomes at 37℃ for 1 hr. Intensities of Alexa568 fluorescence were measured in both samples as $F_{unlabeled}$ and $F_{labelled}$ respectively using the Tecan infinite M1000 microplate reader. FRET, indicating protein-membrane interaction, was observed by the decrease of Alexa568 fluorescence when BimL bound to DiD labeled membranes compared to unlabeled membranes. FRET efficiency was calculated as described previously (*Shamas-Din et al., 2013a*). The data was fit to a binding model as described below. Lines of best fit were calculated using least squares in Graphpad Prism software.

## Calculating the number of Bim molecules per Liposome

Step 1: Find the total number of Bim molecules in a 2 mL reaction

Concentration of Bim: 5 nM in 2 mL reaction

Number of Bim (mole) = concentration (M) x volume (L)

$= 5 \times 10^{-9}$ M x 0.002 L
$= 1 \times 10^{-11}$ mol

Total number of Bim molecules in 2 mL reaction = Number of Bim (mole) x Avogadro's Constant

$= 1 \times 10^{-11}$ mol x $6.02 \times 10^{23}$ mol$^{-1}$
$= 6.02 \times 10^{12}$ Bim molecules

## Step 2: Find the total lipid surface area or the total number of liposome made from 1 mg lipid film

The total number of lipid molecules in a liposome are given by the formula below

$$N_{total} = [4\pi(100\ nm/2)^2 + 4\pi((100\ nm/2)-5)^2]/0.71\ nm^2 = 80088.49\ \text{lipid molecules}$$

Assuming that the average area of the lipid head group in our liposome is 0.71 $nm^2$ (because it is made out of mostly PC).

Total lipid in 1 mg lipid film (in mole)=mass in gram/molecular wt (M.W) in $gmol^{-1}$

= 0.001 g/ 804.95 $gmol^{-1}$
= $1.24 \times 10^{-6}$ mole of lipid

Total lipid molecules in 1 mg lipid film = $1.24 \times 10^{-6}$ mol x $6.02 \times 10^{23}$ $mol^{-1}$

= $7.48 \times 10^{17}$ lipid molecules

Molecular weight of our mitochondrial-like lipid film was calculated from the %Molar and molecular weight of individual lipid which was published in *Kale et al. (2014)* (Examining the Molecular Mechanism of Bcl-2 Family Proteins at Membranes by Fluorescence Spectroscopy).

Total number of liposomes = Total lipid molecule in 1 mg/$N_{total}$ = $9.34 \times 10^{12}$ liposomes.
Surface area of a liposome = $4\pi r^2 = 4\pi(50\ nm)^2 = 10000\pi\ nm^2$
where r is the radius of our liposome.
Total lipid surface area = surface area of a liposome x total number of liposome = $2.93 \times 10^{17}\ nm^2$

## Step 3: Calculate the number of Bim per liposome or per surface area

Number of Bim per liposome = Total number of Bim in a 2 mL reaction/Total number of liposome

= $6.02 \times 10^{12}$ Bim molecules/$9.34 \times 10^{12}$ liposomes=0.64 Bim molecule per liposome

Number of Bim per surface area = Total number of Bim in a 2 mL reaction/Total lipid surface area

= $6.02 \times 10^{12}$ Bim molecules/$2.93 \times 10^{17}\ nm^2$=$2.05 \times 10^{-5}$ Bim molecule per $nm^2$

### Membrane permeabilization

Membrane permeabilization assays with liposomes encapsulating ANTS and DPX were performed as described previously (*Kale et al., 2014*). To measure permeabilization of BMK mitochondria, the indicated amounts of proteins were incubated with mitochondria (1 mg/mL) purified from BMK cells genetically deficient for Bax and Bak expressing mCherry fluorescent protein fused to the SMAC import peptide responsible for localization in the inter-membrane space. After incubation for 45 min at 37°C samples were centrifuged at 13,000 g for 10 min to separate the pellet and supernatant fractions and membrane permeabilization was calculated based on the mCherry fluorescence in each fraction (*Shamas-Din et al., 2014*). For mouse liver mitochondria, cytochrome c release was measured by immunoblotting as described previously (*Pogmore et al., 2016*; *Sarosiek et al., 2013*).

### BH3 profiling

Heavy membranes enriched in mitochondria were isolated as described previously (*Pogmore et al., 2016*; *Brahmbhatt et al., 2016*). Membrane fractions (1 mg/mL) were incubated with 500 nM of the specified BH3-proteins (Bim, Bad and/or Noxa). For E15 brain mitochondria, 0.5 mg/mL of membrane fractions were used and incubated with the indicated amounts of BH3-only proteins for 30 min at 37 °C. Membranes were pelleted by centrifugation at 13,000 g for 10 min and cytochrome c release was analyzed by immunoblotting using a sheep anti-cytochrome c antibody (Capralogics). Mitochondria from embryonic mouse brains for BH3profiling experiments were prepared from ~20 mouse embryos, E15 in age, following the same protocol used for liver mitochondria (*Pogmore et al., 2016*).

### Protein-protein binding

For FRET experiments, single cysteine mutants of cBid (126C), Bcl-XL (152C), Bax (126C), BimL (41C) and BimL mutants were purified and labeled with either Alexa 568-maleimide (donor) or Alexa 647-maleimide (acceptor) as specified. To determine binding a constant amount of donor protein was

incubated with the indicated range of acceptor proteins and where specified liposomes or mitochondria. The intensity of Alexa568 fluorescence with unlabeled or Alexa647-labeled Bcl-XL was measured as $F_{unlabeled}$ or $F_{labeled}$, respectively, and FRET was calculated as described in *Pogmore et al. (2016)*. All measurements were collected using the Tecan infinite M1000 microplate reader. Lines of best fit were calculated using least squares in Graphpad Prism software.

For each pair of proteins a dissociation constant (Kd) was measured in solution and with liposomes. Curves were fit to an advanced function taking into account change of the concentration of acceptor ([A]) when [A] is close to Kd:

$$F = (F_{max}) \left( \frac{([D] + [A] + K_d) - \sqrt{([D] + [A] + K_d)^2 - 4[D][A]}}{2[D]} \right)$$

[D] is the concentration of donor, F indicates the FRET efficiency with the concentration of acceptor as [A], $F_{max}$ is the maximum FRET efficiency in the curve (*Pogmore et al., 2016*).

## Photo and chemical crosslinking of Bim to Bax or Bcl-XL

To produce the proteins for crosslinking using in vitro systems, the DNA sequence encoding murine BimL Cys-null and Lys-null mutant without the His tag and TEV protease recognition site was excised from the pBluescript II KS(+) vector by restriction endonucleases NcoI and ClaI and inserted into the pSPUTK vector (Stratagene, Santa Clara, CA). Mutations as specified in the text were introduced into this sequence using site-directed mutagenesis to generate the single-Lys BimL or single-Cys mutants.

The photocrosslinking method for studying interactions among the Bcl-2 family proteins has been described in detail (*Lin et al., 2019*). Briefly, using the RNAs produced from the single-Lys Bim DNAs in the pSPUTK vector by an in vitro transcription system, [35S]Met-labeled BimL proteins with a single εANB-Lys incorporated at specific locations were synthesized in an in vitro translation system. 10 µL of the resulting BimL proteins were incubated at 37 ˚C for 1 hr with 1 µM of 6H-Bax or 6H-Bcl-XL protein and Bak-/-mouse liver mitochondria (0.5 mg/ml total protein and were resuspended in AT buffer with 80 mM KCl and supplemented with energy regenerating system as described previously *Yamaguchi et al., 2007*) in a 21 µL reaction adjusted by buffer A (110 mM KOAc, 1 mM Mg(OAc)2, 25 mM HEPES, pH 7.5). The mitochondrial and soluble fractions were separated by centrifugation at 13,000 g and 4 ˚C for 5 min, and the mitochondria were resuspended in 21 µL of buffer A. Both mitochondrial and soluble fractions were photolyzed to induce crosslinking via the ANB probe. The resulting samples were adjusted to 250 µL with buffer B (buffer A with 1% Triton X-100 and 10 mM imidazole) and incubated with 10 µL of $Ni^{2+}$-chelating agarose at 4 ˚C for overnight. After washing the $Ni^{2+}$-beads three times with 350 µL of buffer B and one time with 400 µL of PBS, the photoadducts of the radioactive BimL protein and the 6H-tagged Bax or Bcl-XL protein and other proteins bound to the Ni2+-beads were eluted with reducing SDS sample buffer and analyzed by SDS-PAGE and phosphor-imaging.

For chemical crosslinking, [35S]Met-labeled single-Cys or Cys-null BimL and Bax proteins were synthesized from the respective mutant Bim and Bax DNAs in the pSPUTK vector using a transcription/translation (TNT)-coupled in vitro system (Promega, Madison, WI). The resulting BimL and Bax proteins, 2 µL each, were paired as indicated in *Figure 9*, and reduced by 11 µM Tris(2-carboxyethyl) phosphine hydrochloride (TCEP) in a 20 µL reaction adjusted with buffer A at 37 ˚C for 1 hr. The sample was diluted to 110 µL with buffer A and split evenly to two aliquots. For a '60 min' BMH crosslinking reaction, one aliquot was incubated with 0.1 mM BMH and 6 mM EDTA at 25 ˚C for 60 min, and the reaction was stopped by incubation with 50 mM 2-mercaptoethanol at 25 ˚C for 15 min. For a '0 min' control reaction, the other aliquot was incubated with 2-mecaptoethanol and EDTA for 15 min, and then with BMH for 60 min. The resulting samples were precipitated by trichloroacetic acid. The resulting protein pellets were solubilized in reducing SDS sample buffer and analyzed by SDS-PAGE and phosphorimaging.

To obtain the immunoprecipitation data in *Figure 9—figure supplement 1*, the indicated single-Cys BimL and Bax proteins produced by the TNT system, 4 µL each, were paired and reduced by TCEP. The sample was diluted to 260 µL with buffer A and crosslinked by BMH. The resulting sample was divided to two aliquots. The 85 µL or 170 µL aliquot was immunoprecipitated by Bax or Bim

antibody, respectively. Thus, each aliquot was adjusted to 250 µL with IP buffer (100 mM Tris pH 7.5, 100 mM NaCl, 10 mM EDTA, 1 mM PMSF, 1% (v/v) Triton X-100), and received Bax antibody (made in house, 1:1000 dilution) or Bim antibody (Santa Cruz Biotechnology, Dallas, TX, 1:50 dilution). The samples were rotated at 4 ˚C for overnight, and after receiving 25 µL of Protein G Sepharose (50% suspension in IP buffer), rotated for 2 more hours. After centrifugation at 2000 g for 0.5 min, the beads were washed three times with 400 µL of IP buffer and one time with 400 µL of 100 mM Tris pH 7.5 and 100 mM NaCl. The bound proteins were eluted with reducing SDS sample buffer and analyzed by SDS-PAGE and phosphorimaging.

## Measurement of cell death in response to expression of $^V$BimL constructs

HEK293, BMK, MEF, and HCT116 cells were maintained at 37˚C (5% v/v CO2) in dMEM complete [dMEM, 10% Fetal Bovine Serum, 1% essential amino acids (Gibco, Grand Island, NY)]. CAMA-1 were maintained the same environmental conditions but using dMEM/F12 (Gibco, Grand Island, NY). Cell lines were routinely confirmed to be mycoplasma-free using a PCR-based protocol as described by *Hopert et al. (1993)*, and their authenticity was verified by short-tandem repeat (STR) profiling at The Centre for Applied Genomics (Toronto, ON, Canada) for human cells. Murine cell lines have not been authenticated. Cells were seeded in CellCarrier-Ultra 384-well plates (1000 cells/well for BMK and MEF, 2000 cells/well for HEK293 and HCT116, 3000 cells/well for CAMA-1). One day later, cells were transfected using FugeneHD (Promega, Madison, WI) with plasmids encoding Venus, or Venus-fused BimL constructs in an EGFP-C3 backbone. Cell culture medium was added to each reaction (50 µl/0.05 µg DNA) and the whole mix added to each well (50 µl/well) of a pre-aspirated 384-well plate of cells. After 24 hr, cells were stained with Draq5 and Rhodamine-labeled Annexin V and image acquisition was performed using the Opera QEHS confocal microscope (Perkin Elmer, Woodbridge, ON) with a 20x air objective. Untransfected cells and cells treated with 1 µg/mL staurosporine were used as negative and positive controls for Annexin V staining. Cells were identified automatically using software as described previously (*Shamas-Din et al., 2013a*). Intensity features were extracted using a script (dwalab.ca) written for Acapella high content imaging and analysis software (Perkin Elmer, Woodbridge, ON). Cells were scored as Venus or Annexin V positive if the Venus or Annexin V intensity was greater than the average intensity plus two standard deviations for the Venus or Annexin V channels in images of non-transfected cells. Cell death ascribed to the $^V$BimL fusion proteins was quantified as the percentage of Venus-positive cells that were also Annexin V positive. For neuron cultures, cell segmentation using conventional methods could not be achieved due to complex cellular morphologies. Therefore, nuclei were first identified, then a ring region ~10% of nuclear area was drawn around each nucleus. Venus intensity was calculated for this ring region, representing the neuronal cell body, to determine if the neuron was expressing the Venus fluorescent protein.

## Primary neuron cultures

Primary cortical neuron cultures were prepared from embryonic day 15, C57BL/6J mouse embryos as previously described (*Mergenthaler et al., 2012*). All animal breeding and handling were performed in accordance with local regulations and after approval by the Animal Care Committee at Sunnybrook Research Institute, Toronto. Briefly, after separation from hippocampus and subcortical structures, cortices were washed twice with ice-cold PBS, digested with 1x trypsin for 15 min at 37˚C, washed twice with ice-cold PBS and then resuspended with a flame-treated glass pipette in N-Medium (DMEM, 10% v/v FBS, 2 mM L-glutamine, 10 mM Hepes, 45 µM glucose). The dissociated cortices were gently pelleted by centrifugation (200 g for 5 min), N-media was removed, and neurons were resuspended and cultured in Neurobasal-Plus medium (ThermoFisher Scientific) supplemented with B27-Plus (ThermoFisher Scientific) and 1x Glutamax (ThermoFisher Scientific). Neurons were seeded at 5000 cells per well in a 384 well plate (Greiner µclear) after coating with poly-d-lysine (Cultrex). The medium was partially replaced on day 5 in culture with Neurobasal-Plus supplemented with B27-Plus and 1x Glutamax.

Lentivirus to express $^V$BimL and other BimL mutants were cloned into the pTet-O-Ngn2-Puro construct with the Ngn2 gene cut out. This construct was a kind gift from Dr. Philipp Mergenthaler, Charité Universitätsmedizin Berlin. Primary neuron cultures were infected with both $^V$BimL and rtTA

lentiviral particles (~3 µL of each concentrated stock) on the day of seeding. 24 hr later, Neurobasal-Plus medium containing lentiviral particles was removed and replaced with fresh Neurobasal-Plus medium.

Doxycyline (ThermoFisher scientific) was added to 16 day in vitro old cultures of neurons at a concentration of 2 µg/mL to induce VBimL protein expression. 5 hr later, neurons were stained with 5 µM Draq5 (Thermofisher scientific) and 0.1 µM TMRE (Thermofisher scientific), then incubated for 30 min at 37 ˚C. Confocal microscopy was performed immediately after.

## Lentiviral production

Each lentivirus was made using the following protocol adhering to biosafety level two procedures. On day 0, lentiviral vectors psPax2 (10 µg) and pMD2.G (1.25 µg) were mixed with 10 µg of desired VBimL lentiviral construct in 1000 µL of Opti-MEM media (ThermoFisher Scientific). Next, 42 µL of polyethylenimine (PEI) solution [1 mg/mL] was added, the mixture vortexed, then allowed to settle for 15 min at room temperature. After 15 min, $1.5 \times 10^7$ of resuspended HEK293 cells and the transfection solution were mixed and seeded onto a 100 mm culture dish with 10 mL of dMEM complete plus 10 µM of the caspase inhibitor Q-VD-Oph (Selleckchem), and left to incubate at 37 ˚C (5% v/v $CO_2$) for 72 hr. On day 3, media containing lentiviral particles was filter sterilized using a 0.45 µm polyethersulfone filter, and mixed with polyethylene glycol (Bioshop) to achieve a final concentration of 10% (w/v). This was left to mix and precipitate the virus overnight at 4 ˚C. On day 4, the media was centrifuged for 1 hr at 1600 g, supernatant was then removed and the pellet was resuspended with 400 µL of Neurobasal-Plus media (no additives). Resuspended virus was then stored at −80 ˚C until needed.

## Acknowledgements

This work was funded by CIHR grant FRN 12517 to DWA and BL and CIHR Foundation grant FDN143312 to DWA, US NIH grants R01GM062964, OCAST grant HR16-026 and Presbyterian Health Foundation grant to JL, and by an Institutional Development Award from the National Institute of General Medical Sciences of US NIH under grant number P20GM103640. DWA holds the Tier 1 Canada Research Chair in Membrane Biogenesis. QL held a post-doctoral fellowship from the Canadian Breast Cancer Foundation.

## Additional information

### Funding

| Funder | Grant reference number | Author |
|---|---|---|
| Canadian Institutes of Health Research | FRN12517 | David W Andrews |
| Canadian Institutes of Health Research | FDN143312 | David W Andrews |
| National Institutes of Health | R01GM062964 | Jialing Lin |
| National Institutes of Health | P20GM103640 | Jialing Lin |
| Oklahoma Center for the Advancement of Science and Technology | HR16-016 | Jialing Lin |
| Presbyterian Health Foundation | | Jialing Lin |
| Canada Research Chairs | | David W Andrews |
| Canadian Breast Cancer Foundation | | Qian Liu |

The funders had no role in study design, data collection and interpretation, or the decision to submit the work for publication.

## Author contributions
Xiaoke Chi, Dang Nguyen, James M Pemberton, Data curation, Formal analysis, Investigation; Elizabeth J Osterlund, Data curation, Formal analysis, Validation; Qian Liu, Investigation, Methodology; Hetal Brahmbhatt, Data curation, Validation, Methodology; Zhi Zhang, Formal analysis, Investigation; Jialing Lin, Data curation, Formal analysis; Brian Leber, Conceptualization, Methodology; David W Andrews, Conceptualization, Supervision, Funding acquisition, Methodology, Project administration

## Author ORCIDs
Xiaoke Chi https://orcid.org/0000-0002-5269-8389
James M Pemberton https://orcid.org/0000-0001-8386-1081
Jialing Lin http://orcid.org/0000-0002-2126-1571
David W Andrews https://orcid.org/0000-0002-9266-7157

## Ethics
Animal experimentation: All animal breeding and handling was performed in accordance with local regulations and after approval by the Animal Care Committee at Sunnybrook Research Institute, Toronto AUP #18-558.

## Decision letter and Author response
Decision letter https://doi.org/10.7554/eLife.44525.sa1
Author response https://doi.org/10.7554/eLife.44525.sa2

# Additional files

## Supplementary files
• Transparent reporting form

## Data availability
Data generated or analysed during this study are included in the manuscript and supporting files.

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
