## [Decision Letter]

**Acceptance summary:**

The authors discover a novel role of the C-terminus of the BH3-only protein Bim. So far, only the BH3-domain of Bim has been considered crucial for its pro-apoptotic activity because this domain binds to the hydrophobic pocket of BCl^-^2, BCl^-^xL and MCl^-^1 and inhibits their survival action. The C-terminus of Bim was also shown to be important for targeting to the mitochondrial membrane. Now the authors find that the C-terminus of Bim (CTS) has a second function and is required for physiological concentrations of Bim to activate Bax. A mutant of Bim lacking the CTS still inhibited BCl^-^xL but was unable to activate Bax. These data add an important new aspect of the regulation of apoptosis.

**Decision letter after peer review:**

Thank you for submitting your article "The carboxyl-terminal sequence of Bim enables Bax activation and killing of unprimed cells" for consideration by *eLife*. Your article has been reviewed by three peer reviewers, including Volker Dötsch as the Reviewing Editor and Reviewer #1, and the evaluation has been overseen by Philip Cole as the Senior Editor. The following individual involved in review of your submission has agreed to reveal their identity: Christoph Borner (Reviewer #2).

The reviewers have discussed the reviews with one another and the Reviewing Editor has drafted this decision to help you prepare a revised submission.

Essential revisions:

In the reviews and during the discussion, several issues were raised that require further clarification:

1) A more quantitative analysis of the contributions of the C-terminus and the BH3 domain to Bax is necessary. Currently, the results show that without the C-terminus the BH3 domain alone does not bind to Bax. This is correct for the physiological concentrations used. However, to put this research into the framework of all BH3 protein – Bax interactions reported so far, it would be good to show how much of the binding affinity is due to the BH3 and how much is due to the C-terminal domain.

Related to this: In LUVs and isolated mitos, the authors should compare the activity of Bim-dCTS with BH3 Bim, as this would be key to assess the direct activator role.

2) The data provide mainly correlations between mutations of individual amino acids and observed changes in binding/function but without a structural / mechanistic interpretation of how these changes are able to orchestrate the different effects. From a structural point of view it is difficult to see how just a few amino acids can lead to bax binding (without activation), bax activation and binding to the membrane. There are structural models of the Bim BH3 domain binding to bax and a specific model in which this domain binds to the back side of bax has been proposed. How are these new data consistent with these earlier studies? Where are the binding sites of the CTS on bax? On the hydrophobic groove, also on the back side or somewhere completely different? Unfortunately the cross linking data do not provide any information on the binding site on bax. A mechanistic/structural model, for example using NMR titration data (NMR assignments for bax are online available) is required for better understanding these interactions.

3) BimL and in particular Bim-dCTS-Mao contain hydrophobic stretches at their C-terminus (in case of the Mao sequence this is a complete transmembrane helix). This can easily result in major problems due to aggregation and for example loss of binding of the Mao modified sequences in the solution experiments (Figure 7D). The concentrations are in general low (although during overexpression and purification the concentrations are actually quite high), but reach the microM range where aggregation of such peptides is to be expected. How have the authors ensured that no aggregation of the peptides occurs during the binding studies?

4) The interaction between different proteins on membranes are only estimations (or apparent KD as the authors say themselves). These binding measurements include binding of individual proteins to membranes and then with each other. A quantitative analysis of the binding on membranes requires TIRF measurements which take only molecules tethered to membranes into account. The KD should then also have different dimensions (concentration per area).

5) The authors should have shown a BH3 mutant that does not bind to BCl^-^2s as control in Figure 1, to show killing is BH3 dependent in responsive cell lines. They should also include a positive control with Bid in Figures 1 and 2.

---

## [Author Response]

Essential revisions:In the reviews and during the discussion, several issues were raised that require further clarification:1) A more quantitative analysis of the contributions of the C-terminus and the BH3 domain to Bax is necessary. Currently, the results show that without the C-terminus the BH3 domain alone does not bind to Bax. This is correct for the physiological concentrations used. However, to put this research into the framework of all BH3 protein – Bax interactions reported so far, it would be good to show how much of the binding affinity is due to the BH3 and how much is due to the C-terminal domain.Related to this: In LUVs and isolated mitos, the authors should compare the activity of Bim-dCTS with BH3 Bim, as this would be key to assess the direct activator role.

The reviewers raise an interesting point. As data with partial proteins and peptides generally requires μM concentrations and our assays use nanomolar concentrations similar to the concentrations of the proteins in cells, it is difficult to extend our studies to μM concentrations. To contrast our findings with what’s currently known about Bim BH3-Bax protein interaction, we have altered our Discussion, particularly (third paragraph) and make the direct comparison between our results with full-length BimL to data drawn from literature using the BH3-peptide of this protein. In addition, we have extended the x-axes for Figures 3A, B (Increasing the concentration of BimL-dCTS to 3 μM protein) to better compare the activity between BimL and BimL-dCTS. Figure 3A, B demonstrate that BimL-dCTS can directly activate Bax both in our liposome system and purified mitochondria, however, low μM concentrations are required. While this is still less than reported for a Bim BH3-peptide (Data found in Figure 4H – (Sarosiek et al., 2013)) it is so much more than the 10 nM full length protein that performing experiments with higher concentrations of BimL was not useful.

2) The data provide mainly correlations between mutations of individual amino acids and observed changes in binding/function but without a structural / mechanistic interpretation of how these changes are able to orchestrate the different effects. From a structural point of view it is difficult to see how just a few amino acids can lead to bax binding (without activation), bax activation and binding to the membrane. There are structural models of the Bim BH3 domain binding to bax and a specific model in which this domain binds to the back side of bax has been proposed. How are these new data consistent with these earlier studies? Where are the binding sites of the CTS on bax? On the hydrophobic groove, also on the back side or somewhere completely different? Unfortunately the cross linking data do not provide any information on the binding site on bax. A mechanistic/structural model, for example using NMR titration data (NMR assignments for bax are online available) is required for better understanding these interactions.

We agree that NMR studies of full length Bim binding to Bax could be informative. However, to date such experiments have not proven interpretable due to peak broadening in the NMR. Similar effects were reported by the Walensky group necessitating the use of a stapled peptide of low affinity to obtain chemical shift data. Therefore, in order to provide a more structural/mechanistic model for how Bim protein uses both BH3-domain and CTS to activate Bax, we conducted chemical crosslinking experiments and identified a binding site for the Bim CTS in Bax. Our detailed description of the new data can be found in the Results section (–subsection “The Bim CTS binds to the BH3-binding pocket on Bax”) and the Figure 9 and Figure 9—figure supplement 1. Briefly, the chemical crosslinker BMH contains two sulfhydryl reactive moieties separated by a short spacer, and the formation of a BMH-crosslinked Bim-Bax dimer requires a cysteine in Bim that is in close proximity with another cysteine in the interacting Bax. Therefore, a successful crosslink indicates a close proximity between the two cysteines, thereby revealing the Bim binding site in Bax. We validated this approach using a pair of single cysteine Bim and Bax proteins designed to detect the structurally defined BH3-domain/canonical groove interaction (Figure 9A).

Sequence alignment between the Bim BH3 and the CTS revealed that both have the same hydrophobic residues at the h0, h1, h2 and h3 positions, and the same polar or charged residue at the h1+2 or h2+1 positions (Figure 9D). Therefore we investigated binding to the BH3 binding pocket in Bax using Bim with single cysteines in the CTS and Bax with single cysteines in the groove. Indeed, we detected the crosslinked Bim-Bax dimer using these single cysteine proteins (Figure 9A), which suggests that the BH3-binding groove in Bax is also a binding site for the CTS. In addition, we were unsuccessful in attempts at crosslinking the Bim CTS to the back side of Bax. Since these negative data are not interpretable we have not included them in the manuscript.

To determine whether the physical interaction between the Bim CTS or BH3-domain and the Bax groove we detected is functional, we tested the effect of L129E mutation in the CTS or the 4E mutation in the BH3-domain on the crosslinking. The data from these loss-of-function Bim mutants demonstrate that the CTS and BH3-interactions with the groove detected between the Bim and Bax proteins are functionally important for Bim mediated activation of Bax (Figure 9B, C).

3) BimL and in particular Bim-dCTS-Mao contain hydrophobic stretches at their C-terminus (in case of the Mao sequence this is a complete transmembrane helix). This can easily result in major problems due to aggregation and for example loss of binding of the Mao modified sequences in the solution experiments (Figure 7D). The concentrations are in general low (although during overexpression and purification the concentrations are actually quite high), but reach the microM range where aggregation of such peptides is to be expected. How have the authors ensured that no aggregation of the peptides occurs during the binding studies?

Excellent point! Aggregation during the purification of membrane-bound or membrane-binding proteins is always a concern. However, to measure the low concentrations of proteins used in our assays by gel filtration chromatography and western blotting was not feasible. Therefore, we used the Bim proteins labeled with Alexa568. As seen from new Figure 7—figure supplement 2A, at appropriate concentrations the proteins Alexa568 BimL-dCTS and Alexa568 BimL-dCTS-MAO are not aggregating, as they appear as single elution peaks in low molecular weights (Calculated 27 and 17 kDa respectively). Molecular weight calculations from FPLC data can be found in Figure 7—figure supplement 2—source data 1). Indeed BimL-dCTS serves as a control since it does not contain hydrophobic sequences and as an unstructured protein it is difficult to compare its hydrodynamic radius with that of the standard 12 kDa marker cytochrome c. To our surprise Alexa568 BimL eluted as a protein smaller than Alexa568 BimL-dCTS, calculated MW from FPLC: 8 kDa. We speculate that this resulted from some non-specific binding to the chromatography resin as all three proteins are at their correct molecular weights when subjected to SDS-PAGE (Figure 7—figure supplement 2B) or that it is related to the lack of folding of the protein. Consistent with this interpretation, SDS-PAGE demonstrates that all three proteins are not degraded and relatively free of contaminants from purification. This data has been added to the manuscript as Figure 7—figure supplement 2A and B.

In performing these experiments we realized that the some of the assays in Figure 7 use dye labeled BimL and mutants while others used unlabeled BimL. To remain consistent within the figure and the figure supplements we replaced the data presented in (old) Figure 7B and E which used non-labelled BimL-dCTS-MAO with data for Alexa568 BimL-dCTS-MAO. This protein appears monomeric as shown in Figure 7—figure supplements 2A-B.

4) The interaction between different proteins on membranes are only estimations (or apparent KD as the authors say themselves). These binding measurements include binding of individual proteins to membranes and then with each other. A quantitative analysis of the binding on membranes requires TIRF measurements which take only molecules tethered to membranes into account. The KD should then also have different dimensions (concentration per area).

The reviewers are absolutely correct, the Kd’s we present when measuring the interactions between Bim and Bax, (or Bim and BCl^-^XL) are “apparent” Kds, since we cannot quantify the fraction of the protein complexes that occur on the membranes versus in solution. In order to understand the contribution of the Bim CTS in the binding interaction with Bax taking place on the membrane, our experiments cannot be done with TIRF for BimL-dCTS as this mutant can no longer bind to membranes. Moreover, having published ourselves methods equivalent to TIRF we are not convinced that the supported bilayer is a good model system due to changes in the fluidity of the supported side and the number of different species that need to be identified and characterized. Even with supported bilayers we see fast moving molecules that are unlikely attached to the membrane (Kurylowicz et al., 2013; DOI: https://doi.org/10.1016/j.bpj.2012.11.1251). Finally curvature is also a factor with Bax preferentially inserting into curved membranes.

To ensure all Alexa568 labelled Bim proteins were bound to the membrane surface, both BimL and BimL-dCTS-MAO were incubated with mitochondrial-like liposomes, then passed through a size-exclusion column. Thus, the Bim protein that elutes in the same fraction as the liposomes is the membrane-bound Bim protein. Bim protein that remains in solution is collected in later fractions that is excluded from the experiment. Next, to these isolated liposomes (of defined size, 100nm diameter spheres, and therefore defined area) increasing amounts of acceptor labelled Bax protein was added. FRET was measured from the decreasing fluorescence signal from the donor-labelled (membrane bound) Bim. This data has been used to create the new Figure 7B and 7D. The experiments are described in the last two paragraphs of the subsection “Different residues in the Bim CTS regulate membrane binding and Bax activation”.

Measuring the concentration of both Bim proteins (via Alexa568 fluorescence), and knowing the concentration of liposomes (with a diameter of 100nm), we were able to calculate that there are 0.6 Bim molecules per liposome in our assay. This value corresponds well with the fraction of the liposomes that are permeabilized (Figure 7B). Because our assay contains less than 1-Bim molecule per liposome, this strongly suggests that in the presence of liposomes the BimL-dCTS-MAO protein has not aggregated. This is now described in the aforementioned paragraphs and Materials and methods (subsection “Bim binding to membranes”).

In these experiments with liposomes isolated with bound Bim we still add soluble Bax therefore, the Bim-liposome complex and Bax are both in solution. That makes calculating 2 dimensional Kd’s very complicated. Moreover, in cells approximately half of the Bax is cytoplasmic with the remainder partitioned between molecules peripherally and integrally bound to the membrane. In our opinion given the compromises inherent in all model systems the liposome/mitochondria systems are the most representative of the interactions taking place in cells.

5) The authors should have shown a BH3 mutant that does not bind to BCl^-^2s as control in Figure 1, to show killing is BH3 dependent in responsive cell lines. They should also include a positive control with Bid in Figures 1 and 2.

BimL-4E is a mutant of BimL in which the four hydrophobic residues within the BH3-domain have been mutated to glutamic acid that we have published previously does not bind BCl^-^2 family proteins. Here we show that when expressed in cells, this mutant does not kill (Figure 1A, 2A, 2B), which can be explained by its inability to bind Bax or BCl^-^XL from our in vitro data (Figure 6, Figure 6—figure supplement 1). We apologize for not properly highlighting this important control that was included in the original paper and have changed the text to reflect this (subsection “The CTS of Bim variably contributes to the pro-apoptotic activity of Bim in different cell lines”, first paragraph).

As suggested by the reviewers we added experiments (Figure 1 and 2) including the positive control tBid that is N-terminally fused with the fluorescent protein Venus (^V^tBid) (see the second paragraph of the aforementioned subsection).